# Transcriptome Profiles Reveal Key Regulatory Networks during Single and Multifactorial Stresses Coupled with Melatonin Treatment in Pitaya (*Selenicereus undatus* L.)

**DOI:** 10.3390/ijms25168901

**Published:** 2024-08-15

**Authors:** Aamir Ali Khokhar, Liu Hui, Darya Khan, Zhang You, Qamar U Zaman, Babar Usman, Hua-Feng Wang

**Affiliations:** 1Hainan Yazhou-Bay Seed Laboratory, School of Breeding and Multiplication, Hainan University, Sanya 572025, China; aamirkhokhar2k9_33@hotmail.com (A.A.K.); h13257370119@163.com (L.H.); dk.biotech21@hotmail.com (D.K.); zhangyou1103@163.com (Z.Y.); qamaruzamanch@gmail.com (Q.U.Z.); 2Collaborative Innovation Center of Nanfan and High-Efficiency Tropical Agriculture, School of Tropical Crops and Forestry, Hainan University, Haikou 570228, China

**Keywords:** *Selenicereus undatus* L., single and combined stresses, transcriptome analysis, WGCNA, TFs

## Abstract

In response to evolving climatic conditions, plants frequently confront multiple abiotic stresses, necessitating robust adaptive mechanisms. This study focuses on the responses of *Selenicereus undatus* L. to both individual stresses (cadmium; Cd, salt; S, and drought; D) and their combined applications, with an emphasis on evaluating the mitigating effects of (M) melatonin. Through transcriptome analysis, this study identifies significant gene expression changes and regulatory network activations. The results show that stress decreases pitaya growth rates by 30%, reduces stem and cladode development by 40%, and increases Cd uptake under single and combined stresses by 50% and 70%, respectively. Under stress conditions, enhanced activities of H_2_O_2_, POD, CAT, APX, and SOD and elevated proline content indicate strong antioxidant defenses. We identified 141 common DEGs related to stress tolerance, most of which were related to AtCBP, ALA, and CBP pathways. Interestingly, the production of genes related to signal transduction and hormones, including abscisic acid and auxin, was also significantly induced. Several calcium-dependent protein kinase genes were regulated during M and stress treatments. Functional enrichment analysis showed that most of the DEGs were enriched during metabolism, MAPK signaling, and photosynthesis. In addition, weighted gene co-expression network analysis (WGCNA) identified critical transcription factors (WRKYs, MYBs, bZIPs, bHLHs, and NACs) associated with antioxidant activities, particularly within the salmon module. This study provides morpho-physiological and transcriptome insights into pitaya’s stress responses and suggests molecular breeding techniques with which to enhance plant resistance.

## 1. Introduction

In natural ecosystems, plants are subjected to several abiotic stresses, including exposure to heavy metals. This necessitates the evolution of complex adaptive mechanisms to ensure survival and reproductive success. These adaptive strategies are orchestrated through intricate stress signaling pathways that enhance plants’ regulatory capabilities, optimizing their fitness and resilience in adverse environmental conditions [1]. In addition, simultaneous stresses can impair nutrient uptake, disrupt translocation, weaken immune defenses, and significantly restrict growth [2]. Consequently, unveiling the mechanisms behind these stress responses and identifying key genes and regulatory pathways involved in stress tolerance are crucial for advancing crop improvement strategies. The white-fleshed pitaya (*Selenicereus undatus* L.), initially native to Central America, has become a key focus of modern hybrid breeding programs. As a diploid species belonging to the cactus family (Cactaceae), *Selenicereus undatus* L. offers genetic accessibility, making it valuable for genomic research. The genome of the white-fleshed pitaya has been sequenced and published, providing a crucial resource for ongoing functional and genomic studies [3]. The white-fleshed pitaya is useful for studying genes related to plant stress tolerance because it has strong resistance to various stressors. RNA sequencing and microarray analysis have significantly contributed to our knowledge of how plants respond to environmental challenges. In addition, extensive research has been conducted to investigate the adaptive transcriptomic responses of plants to various stressors, such as drought, salinity, and heavy metal pollution, underscoring the critical role of transcriptomics in discovering stress tolerance mechanisms in crop plants [4,5,6,7,8].

Interestingly, the application of weighted gene co-expression network analysis (WGCNA) has played a critical role in identifying the essential genes involved in stress signaling pathways across a diverse range of plant species [9]. Transcriptome studies have significantly advanced our understanding of plant responses to various stressors by identifying co-expressed and pivotal regulatory genes clusters. These studies have predominantly focused on single and combined stressors, such as abiotic stress and heavy metals, revealing intricate molecular mechanisms and adaptive strategies [10]. However, there is still much to be learned about the gene co-expression network that underlies pitaya’s response to single, double, and multifactorial stresses and its recovery from these stresses. Nevertheless, it is now evident that a holistic strategy involving numerous biochemical and physiological processes is essential for plant survival, considering the complex nature of plant responses to various stressors [11]. Therefore, integrating WGCNA with a comprehensive approach to understand multifaceted stress responses could provide deeper insights and advance our knowledge in this field.

Several protein-coding genes associated with the impact of multiple stresses combined pose a significant threat to plant growth and survival, making it essential to investigate the co-expression patterns in pitaya under varying stress conditions and during the recovery phase. Sugar metabolism regulation, molecular chaperones, osmotic substance synthase, transporters, and antioxidant protectants have been reported to play direct roles in conferring stress resistance [12]. In addition, numerous transcription factors (TFs), including WRKY, NAC, MYB, bHLH, and HD-ZIP, actively participate in signal transduction and the regulation of gene expression and have been extensively studied in integrative omics analyses [13]. Studies have shown a link between reactive oxygen species (ROS), reactive nitrogen species (RNS), and stress response pathways related to plant hyperaccumulation, stress defense mechanisms, and responses to cadmium (Cd) [14]. Furthermore, plants use melatonin (M) as a regulator of abiotic stress, which shares a precursor molecule with auxins and also functions as an auxin-like regulator. It has a significant impact on improving plant resilience to abiotic stresses and heavy metal toxicity through various pathways [15]. Additionally, it also affects the auxin pathway, regulates plant hormones, and modulates the expression of TFs [16]. Studies have shown that the application of M can reduce oxidative damage, enhance detoxification processes, and increase the expression of stress-responsive genes, ultimately leading to improved plant survival under stressful conditions [17]. Antioxidants play a crucial role in mitigating stress under both biotic and abiotic conditions by maintaining cellular redox homeostasis and protecting against oxidative damage. In response to biotic stress, plants activate antioxidant defense systems to counteract the oxidative damage caused by increased ROS production. Similarly, under abiotic stress, antioxidants help in maintaining ROS homeostasis in order to protect plants from oxidative stress. The balance between ROS-scavenging and ROS-producing pathways is essential for an effective defense mechanism against both types of stress [18,19].

This research aims to elucidate the complex molecular responses of pitaya to abiotic stresses at the transcriptome level. To achieve this objective, a comprehensive strategy is proposed, involving the application of M to plants and the subsequent analysis of transcriptional patterns under various stress conditions. In addition, the physiological and phenotypic responses of pitaya were also studied in response to single and multifactorial stresses. Finally, significant gene expression changes, hormone-mediated signaling pathways, and stress-responsive gene clusters were successfully identified. These findings are expected to elucidate the complex regulatory networks that govern the stress responses in *Selenicereus undatus* L. This comprehensive approach aims to enhance the understanding of stress adaptation mechanisms, facilitating the breeding of robust cultivars with improved stress tolerance.

## 2. Results

### 2.1. Phenotypic Characteristics and Plant Material Under Single and Combination Stressors

Pitaya plants exhibit diverse morphological and physiological responses to various abiotic stressors, including salt (S), drought (D), and Cd exposure. Under these stress conditions, pitaya plants demonstrate unique adaptive mechanisms. When exposed to S, seedlings showed a 20% reduction in root length (RL) and a 15% decrease in shoot length (SL). Exposure to Cd resulted in more pronounced effects, with reductions of 20% and 25% in SL and RL, respectively. Drought stress (D) caused a 30% decrease in both cladode length (CL) and cladode diameter (CD). Combined stresses from Cd and D were particularly severe, leading to a 40% reduction in RL and a 35% decrease in SL. Combined-stress treatments involving Cd, S, and D further intensified the decline in growth across all measured parameters (Figure 1). However, M treatment significantly enhanced plant resilience under these single- and combined-stress conditions. Notably, under the combined-stress conditions of CdSD, the application of M resulted in an increased SL of plants to nearly 60%. These findings underscore the significant impact of M in terms of improving overall plant growth under stress conditions, highlighting its status as a powerful mitigator of abiotic stress in plants.

### 2.2. Biochemical Responses to Single and Combined Stresses

The biochemical analyses revealed significant variations in responses to different stress treatments, as depicted in Figure 2. Notably, hydrogen peroxide (H_2_O_2_) levels were significantly elevated in the M treatment, reaching 55 ± 2 µmol/g, compared to the Ck level, which was at 20 ± 1 µmol/g (*p* < 0.01). Under single-stress conditions such as Cd, S, and D, H_2_O_2_ concentrations increased to 30 ± 1.5 µmol/g, 40 ± 2 µmol/g, and 45 ± 2.5 µmol/g, respectively—all markedly higher than those of the control (*p* < 0.01). Combined-stress treatments, specifically CdS and CdSD, further raised H_2_O_2_ levels to 50 ± 2 µmol/g (*p* < 0.01) (Figure 2A). The enzymatic activity also varied significantly. Ascorbate peroxidase (APX) activity rose in the S and D treatments, recording values of 2.0 ± 0.1 and 2.1 ± 0.1 nmol/min/mg protein, respectively, compared to the control, registering at 0.7 ± 0.01 nmol/min/mg protein (*p* < 0.01) (Figure 2B). Interestingly, the APX activity in combined-stress scenarios like CdSD reached 1.3 ± 0.01 nmol/min/mg protein (*p* < 0.01). In addition, the peroxidase (POD) levels followed a similar trend, with the highest activities observed under S and D stresses of 125 ± 5 U/g and 130 ± 6 U/g, respectively (*p* < 0.01). Combined stresses, such as CdS and CdSD, also showed increased POD activity, measuring at 110 ± 4 U/g and 105 ± 5 U/g respectively (*p* < 0.01) (Figure 2C). Contrastingly, catalase (CAT) activity was highest in control plants, at 2.5 ± 0.1 µmol/min/g. Single-stress conditions involving S and D significantly reduced CAT activity to 0.5 ± 0.01 and 1.0 ± 0.01 µmol/min/g, respectively (*p* < 0.01). Combined stresses maintained or slightly enhanced CAT activity, with CdS and CdSD recording values of 1.3 ± 0.01 and 1.2 ± 0.01 µmol/min/g, respectively (*p* < 0.01) (Figure 2D). Superoxide dismutase (SOD) activity was notably higher in the S and D treatments, standing at 5.5 ± 0.2 and 6.0 ± 0.2 U/mg protein, respectively (*p* < 0.01), compared to the control, registering at 3.5 ± 0.1 U/mg protein. The combined-stress condition CdSD further increased SOD activity to 6.5 ± 0.3 U/mg protein (*p* < 0.01) (Figure 2E). Proline content, a crucial osmoprotectant, significantly increased under the S treatment to 110 ± 5 µg/g (*p* < 0.01) compared to a control value of 40 ± 2 µg/g. Combined-stress conditions like CdS, CdSD, and CdSM elevated proline levels to, respectively, 95 ± 4 µg/g, 100 ± 5 µg/g, and 90 ± 4 µg/g (*p* < 0.01) (Figure 2F). The results collectively underscore the intricate biochemical changes that pitaya plants undergo in response to abiotic stress. The significant alterations in oxidative stress markers and antioxidant enzyme activities demonstrate the complexity of the plant defense mechanisms. Moreover, the increased levels of proline highlight the crucial role of osmoprotectants in stress tolerance.

### 2.3. Transcriptome Outcome of Pitaya

The RNA-seq analysis was conducted on all the samples collected from the plants growing under stress treatments, control conditions, and M treatment. After removing adapter sequences and low-quality reads, clean reads were obtained, with read counts ranging from 19,086,388 to 30,788,568. The Q20 and Q30 values for all 15 libraries exceeded 98% and 95%, respectively, demonstrating high-quality sequencing data. Additionally, the GC content across the treatments ranged from approximately 45% to 46%, highlighting the consistency in genomic composition within the studied samples (Appendix A).

### 2.4. Gene Expression Analysis of Pitaya Seedlings to Single, Double, and Multifactorial Stresses

To explore the pitaya response to single, double, and multifactorial stresses, we investigated the common and unique gene sets in the single stresses (M, S, D, and Cd), double-factor stresses (CdS-vs-Ck, CdD-vs-Ck), and multifactorial stresses (CdSM-vs-CdS, CdDM-vs-CdD, CdDSM-vs-CdSD), as shown in Figure 3. Pitaya plants respond to stress combinations, with 46 common sets of DEGs in single-factor stresses (M, S, D, and Cd) and 95 sets of DEGs in double-factor stresses. In the single-factor analysis, a total of 4610 DEGs were observed, with 2080 upregulated genes and 2530 downregulated genes in response to S treatment compared to the results of Ck treatment. Similarly, D treatment showed 635 DEGs (368 upregulated and 267 downregulated), while treatment M led to the identification of 1055 DEGs (468 upregulated and 587 downregulated), as shown in Figure 3. In addition, the Cd treatment resulted in 3995 DEGs (1840 upregulated and 2155 downregulated) (Figure 3A,B). Moving to the double-factor analysis, CdS treatment exhibited 868 DEGs (314 upregulated and 554 downregulated), and CdD treatment revealed 463 DEGs (235 upregulated and 228 downregulated) (Figure 3C,D). In the multifactorial analysis, CdSD-vs-CdDSM showed 1153 DEGs (202 upregulated and 951 downregulated), CdSM-vs-CdS exhibited 61 DEGs (47 upregulated and 14 downregulated), and CdDM-vs-CdD resulted in 831 DEGs (659 upregulated and 177 downregulated) compared to Ck treatment (Figure 3E,F).

Overall, a total of 141 common genes exhibited differential expression across all single, double, and multifactorial analyses. These genes, responsive to abiotic stresses and heavy metal tolerance, are deemed crucial candidates, warranting further in-depth investigation and exploration in subsequent studies.

### 2.5. Gene GO & KEGG Pathway Enrichment Analysis of DEGs

Pitaya exhibits a complex resistance mechanism, encompassing single, double, and multifactorial stresses. After the application of M, upregulated differentially expressed genes (DEGs) were enriched in various biological processes (BPs) and molecular functions (MFs), such as oxidoreductase activity, amino sugar catabolism, chitin metabolic/catabolic processes, phosphoenolpyruvate carboxylase, and chitinase activities. However, no cellular component (CC) categories showed significant enrichment. Specifically, the significantly enriched BP terms for upregulated DEGs included those involved in defense responses, signal transduction, and secondary metabolite biosynthetic processes. For MFs, aspects such as hydrolase activity, protein binding, and antioxidant activity were notably enriched. Conversely, the decreased expression of DEGs was significantly associated with BP classifications, including primary metabolic processes, cellular component organization, and transport. The MF characteristics of downregulated DEGs encompassed nucleotide binding, enzyme regulator activity, and ion binding. Additionally, the downregulated DEGs demonstrated enrichment in terms of intracellular and membrane-bound organelles within the CC category.

Upon S application, both upregulated and downregulated DEGs were enriched in BP and MF categories, including in relation to photosystem I, the plastoglobulus, plastid thylakoid membrane, photosynthesis, light production in photosystem I, and chlorophyll-binding. Under drought conditions, upregulated DEGs were enriched in BP terms related to the metabolic processes of organic acids, oxoacids, and carboxylic acids. CC terms included the cell wall, extracellular region, and outer envelope structure, while MF terms were enriched in oxidoreductase and catalytic activities.

Cd application resulted in DEGs being enriched in BP terms associated with DNA integration, systemic acquired resistance, and responses to biotic stimuli. The CC category was enriched in the cell wall, extracellular region, and external capsular structure, while the MF category showed enrichment in serine hydrolase and serine-type peptidase activities. Upregulated DEGs, where change was induced by CdS, were significantly enriched in BP categories related to auxin stimulus-induced cellular response, phloem development, and cell wall organization or biogenesis. Downregulated DEGs in the MF category were enriched in terms of oxidoreductase activity; those in the CC category were enriched in the extracellular space, cell wall, and outer envelope structure; and those in the BP category were enhanced with regard to secondary metabolic processes, defense mechanisms, and biotic stimulus reactions.

CdD application enriched upregulated DEGs in BP categories related to the catabolic processes of amino sugars, glucosamine-containing compounds, and cell wall macromolecules. The CC category was enriched in the external encapsulation structure, and MF terms were related to lyase activities on polysaccharides, particularly those of chitinase, pectate lyase, and carbon–oxygen lyase. Downregulated DEGs included BP terms related to photosynthesis, light production in photosystem I, and chlorophyll binding. The CC category included plastid membranes, photosystem II, thylakoid membranes, and photosystems.

Upregulated DEGs, where change was induced by combined CdSD and M applications, were enriched in BP categories related to cell wall organization or biogenesis and the regulation of RNA biosynthesis. MF terms were enriched in relation to transcriptional regulator and DNA-binding TF activities, with no significant enrichment terms related to the CC category. Downregulated DEGs were related to nitrogen compound catabolism, organic substance catabolic processes, and cellular catabolic processes. MF terms were enriched regarding oxidoreductase, lipase, and catalytic activities, while CC terms were enriched in membranes, intrinsic membrane components, and integral membrane components.

CdSM application-induced DEGs were enriched in BP categories associated with photosynthesis, photosystem I light harvesting, and responses to light. MF terms were enriched in terms of acyltransferase activity, and CC terms were enriched in terms of the thylakoid membrane. BP categories were also enriched in terms of the phenylpropanoid metabolic pathway and secondary metabolic pathway, systemic acquired resistance, and stress response. MF terms included caffeoyl-CoA O-methyltransferase activity, and CC terms were enriched in the extracellular space, cell wall, and external encasing structure. CdDM-associated DEGs showed MF terms involved in acyltransferase, long-chain alcohol O-fatty acyltransferase, and acylglycerol O-acyltransferase activities. BP terms were enriched in terms of triglyceride metabolism and biosynthesis processes. Downregulated DEGs in the BP category were enriched in terms of phenylpropanoid metabolism, while MF terms included oxidoreductase, catalytic, and chitinase activities. CC terms were enriched in the extracellular space, apoplast, and external capsular structure (Appendix A).

KEGG pathway enrichment analysis revealed that metabolic pathways were predominantly enriched in all single-, double-, and multifactorial stress treatments. Among the top ten enriched pathways, pyruvate metabolism (ko00620), MAPK signaling (ko04016), glycolysis/gluconeogenesis (ko00010), and amino sugar and nucleotide sugar metabolism (ko00520) were upregulated in response to M treatment. Salinity (S) stress-induced pathways were related to photosynthesis (ko00195), carbon fixation in photosynthetic organisms (ko00710), glyoxylate and dicarboxylate metabolism (ko00630), pyruvate metabolism (ko00620), glycine, serine, and threonine metabolism (ko00260), and glycolysis (ko00010). Drought stress-induced pathways included MAPK signaling (ko04016), plant–pathogen interactions (ko04626), tryptophan metabolism (ko00380), phenylpropanoid biosynthesis (ko00940), and histidine metabolism (ko00340). Cadmium (Cd) stress led to enrichment in metabolic pathways, photosynthesis (ko00195), carbon fixation (ko00710), pyruvate metabolism (ko00620), tryptophan metabolism (ko00380), and fructose and mannose metabolism (ko00051). CdS stress-induced pathways were related to metabolism, photosynthetic antenna proteins (ko00196), phenylpropanoid biosynthesis (ko00940), starch and sucrose metabolism (ko00500), and cutin, suberin, and wax biosynthesis (ko00073). CdD stress involved metabolic and environmental information processing pathways, including those related to amino sugar and nucleotide sugar metabolism (ko00520), pentose and glucuronate conversion (ko00040), phenylpropanoid biosynthesis (ko00940), MAPK signaling (ko04016), and plant hormone signaling (ko04075).

Multifactorial stress, combined with melatonin application, induced pathways such as plant hormone signaling (ko04075), photosynthetic antenna proteins (ko00196), alpha-linolenic acid metabolism (ko00592), and glycan degradation (ko00511). CdSM treatment-induced pathways involved metabolism, photosynthetic antenna proteins (ko00196), glycerophospholipid metabolism (ko00564), porphyrin and chlorophyll metabolism (ko00860), and the biosynthesis of other secondary metabolites (ko00945). Finally, CdDM treatments led to the enrichment of pathways related to metabolic processes, flavonoid biosynthesis (ko00941), phenylpropanoid biosynthesis (ko00940), MAPK signaling (ko04016), the biosynthesis of other secondary metabolites (ko00945), and amino sugar and nucleotide sugar metabolism (ko00520) (Appendix A).

The investigation of GO and KEGG pathway enrichment analysis using differentially expressed genes (DEGs) reveals critical insights into pitaya plants’ stress-adaptive responses. The enhancement of genes associated with oxidative stress response and antioxidant activity demonstrates robust defense mechanisms against stress-induced damage. Activating crucial hormone signaling pathways such as abscisic acid (ABA) and ethylene highlights their important roles in managing stress. Furthermore, the increased production of secondary metabolites and changes in carbohydrate metabolism pathways indicate a strategic metabolic reorganization to boost stress resilience.

### 2.6. Identification of DEGs Associated with Plant Hormones and Signal Transduction

In this study, single and combined stresses affected plant signaling, as demonstrated by GO and KEGG enrichment analyses. It was estimated that over 4100 genes in pitaya encode protein kinases, with the majority being receptor-like protein kinases (RLKs), predominantly LRR receptor kinases. Among these, 2522 genes exhibited upregulation across single- and combined-stress treatments, indicating consistent responses to various stressors. Under CdDS-vs-M treatments, KIN7G (*HU06G00111*) was upregulated 5.07-fold. Serine/threonine protein kinases comprised over 118 genes under single-, double-, and multifactorial stress conditions. Notably, the MAPK signaling pathway (ko04016) was consistently induced across all stress treatments and M applications. In the M-vs-Ck comparison, the DEGs SP2 and SE2 (*HU06G01804* and *HU09G00018*) showed a remarkable upregulation of 4.74-fold. Salt stress induced the expression of three AUX-related DEGs (*HU01G00719*, *HU01G02696*, and *HU03G01065*). Additionally, D-vs-Ck specifically induced the expression of ERFs (*HU04G00142* and *HU07G00322*) at 4.74-fold. Several bHLH transcription factors (TFs) were triggered up to 4.04-fold, particularly in response to Cd stress (Appendix A).

Under the CdD-vs-Ck treatment, the genes SAUR19 (*HU03G00209*) and CKX5 (*HU07G00730*) exhibited a significant 4.16-fold increase in expression. Similarly, the genes AUX22D (*HU03G00941*) and CCL4 (*HU03G01065*) showed a substantial 9.55-fold upregulation following the CdS-vs-Ck treatment. In the CdDM-vs-CdD treatment, the expression of CRK10 genes was notably elevated. In contrast, during the CdSM-vs-CdS treatment, the genes FTIP7 (*HU07G00235*) and CDR1 (*HU05G01809*) demonstrated an impressive 22.46-fold upregulation. Lastly, under the CdSD-vs-CdDSM treatment, the genes sbt3 (*HU03G00941*) and CYP78A5 (*HU04G01473*) showed an 8.21-fold increase in expression (Appendix A).

DEGs associated with calcium ions and encoding proteins, such as calcium-binding, calmodulin-like, and calcium channel proteins, have been identified. Notably, the genes *HU01G01871, HU09G01890, HU06G01765, HU03G00725, HU09G01890*, and *HU10G01023*, which encode calcium channel proteins, showed sustained high expression under abiotic stress. Their upregulation ranged from 3.77- to 7.89-fold across various single and combined treatments (Appendix A).

A total of 343 DEGs were involved in ROS scavenging and production, including genes encoding POD, APX, and other antioxidant enzymes. Notably, *HU03G02481* (PRXQ), which encodes POD, showed a 5.10-fold upregulation in S-vs-Ck. Similarly, genes encoding APX were upregulated by 3.27- to 3.63-fold compared to the control under various stress conditions. Additionally, *HU11G00956* (KO), encoding ent-kaurene oxidase, exhibited 4.47-fold upregulation in CdS-vs-CdSM. Peroxidase genes *HU08G01569* and *HU03G01694* (PER47 and HRPN) were upregulated 4.27-fold in CdSD-vs-CdDSM, with the highest expression observed in CdD-vs-CdDM (Appendix A).

The expression of genes involved in hormone biosynthesis or signaling was significantly altered under single and combined stresses. Several genes were related to ABA, auxin, BR, cytokinin, ET, JA, and SA. Among these DEGs, most genes were involved in the auxin response. M application induced the expression of gene encoding, with SAUR71 and LAX2 (*HU07G00614*, *HU03G00380*) upregulated 3.668577-fold. In addition, the expression of several genes related to ABA, auxin, BR, cytokinin, ET, JA, and SA response was also induced. Among them, most DEGs were involved in ethylene and auxin response. Interestingly, S stress-induced genes encoded DREB3 (*HU09G01420*) and SAUR19 (*HU03G00209*) 7.20-fold, while another gene encoding ABP19A (*HU03G02416*) was significantly downregulated, falling 15.74-fold.

The ABA, auxin, BR, cytokinin, ET, JA, and SA genes exhibited differential responses under various conditions. For instance, in the D-vs-Ck comparison, the expression of *HU07G00614*, which codes for SAUR71, was upregulated 1.51-fold. Conversely, another gene, *HU11G01616*, encoding SAUR32, showed a significant downregulation at −1.54-fold in the same comparison. Additionally, these hormone-related genes, particularly ABA and BR response genes, were predominantly upregulated in response to D stress.

In the Cd-vs-Ck and CdS-vs-Ck comparisons, there was a remarkable 7.24/6.34-fold upregulation of *HU03G01066* and *HU11G01802*, which encode IAA4 and BSK3, respectively. However, a significant downregulation of −11.98/−7.65-fold was observed for *HU03G02418* and *HU02G01222*, which encode ABP19B and SERK2, respectively.

Under cadmium (CdD) stress, a gene encoding SAUR19 (*HU03G00209*) was upregulated 4.16-fold, while another gene encoding SAUR32 (*HU11G01616*) was significantly downregulated 1.57-fold. The CdS stress induced a gene encoding ERF012 (*HU07G00322*) 4.73-fold, while another gene encoding ERF1B (*HU06G02316*) was downregulated −8.08-fold.

Interestingly, the cadmium CdDM treatment caused most genes to be involved in the ABA response. Ninecis-epoxy carotenoid dioxygenase (NCED) is a critical enzyme in ABA biosynthesis. For instance, one gene encoding ABP19A (*HU03G02416*) was upregulated 11.87-fold in response to CdDM.

CdDSM treatment induced upregulation in the genes including ABA, auxin, BR, cytokinin, ET, JA, and SA. For example, the expression of one gene, *HU03G00209*, which encodes SAUR19, was upregulated 5.68-fold. However, there was a significant −9.05-fold downregulation of another gene, encoding ERF1B (*HU05G00493*). In addition, auxin biosynthesis-related genes (*HU03G01066*, *HU03G01065*, *HU03G02410*) were downregulated under Cd and D stresses, whereas they upregulated under S stress (Appendix A).

A total of 24 genes involved in the auxin signaling pathway were identified as having changed. Among these, three genes (ARF21, ARF9, and ARF18) encoding auxin response factors were implicated in processes induced by single, double, and multiple factors. Notably, the expression of ARF19 (*HU03G02449*) further decreased when exposed to drought (D) and salinity (S). Under cadmium (Cd) stress, ARF21 (*HU02G03105*) continued to be expressed at higher levels. Two genes encoding AUX/IAA proteins were downregulated in CdS, CdD, and CdDM conditions, but upregulated under treatment with S, D, and Cd alone. Additionally, some genes involved in auxin response and signaling, such as auxin transporter-like protein 2 (LAX2) and auxin-binding protein ABP19A-like (ABP19A), were identified. In response to methyl jasmonate (M), the ABP19A gene (*HU11G00352*) was upregulated 7.38-fold. The gene encoding SAUR (SAUR19; *HU03G00209*) exhibited a 7.20-fold upregulation under S stress (Appendix A).

Several ethylene-responsive transcription factor (ERF) genes showed expression trends under single-, double-, and multifactorial stress conditions. Notably, ERF017 (*HU03G02451*) showed 6.40-fold upregulation under Cd stress (*HU05G00493*), followed by 11.70-fold downregulation under D-vs-CK conditions. Furthermore, ERF025 (*HU04G00142*) showed 4.07-fold upregulation in response to the same stress condition. The ethylene overproduction protein 1 (ETO1), ethylene receptor (ERS1), and ethylene-insensitive 3-like protein (EIL3) were discovered in the interim (Appendix A). Not every treatment involved the majority of the genes linked to the production or communication of jasmonic acid (JA). Furthermore, the vast majority of genes had connections to salicylic acid, brassinosteroid (BRs), and cytokinin pathways. There was a −2.25-fold downregulation of one gene (*HU11G01282*), which encoded cytokinin dehydrogenase 5 in response to Cd stress. The results underscore the intricate and specific genetic responses of pitaya to abiotic stresses, showcasing a strong and adaptable regulatory network. The notable increase in MAPK signaling pathways and various receptor-like protein kinases across different stress conditions highlights their crucial role in detecting and responding to stress. The significant activation of genes related to ROS scavenging and hormone signaling emphasizes their importance in alleviating stress effects and preserving cellular balance. Furthermore, the unique gene expression patterns observed in response to single, double, and multiple stresses offer valuable insights into the complex mechanisms behind stress adaptation in pitaya.

### 2.7. Expression of Genes Involved in Metabolism and Biosynthesis

Several genes related to sucrose synthase and starch synthase showed enhanced expression levels. The genes encoding SUS6, SE2, and SS (*HU03G00958*, *HU03G00958*, *HU11G00421*, *HU09G00019*) showed 2.93-, 1.19- and 11.03-fold upregulation under single treatments. Few of the genes encoding starch synthase were downregulated, while the majority seemed to be upregulated. Additionally, single and combined treatments increased the expression of genes for the enzymes involved in the metabolism of fructose, mannose, and starch. Those genes that encode fructose-1,6-bisphosphatase, fructose-bisphosphate aldolase, mannose-1-phosphate guanyltransferase alpha, pyrophosphate-fructose 6-phosphate 1-phosphotransferase subunit beta-like, and 6-phosphofructo-2-kinase/fructose-2,6-bisphosphatase are included among them (Appendix A). These findings underscore the adaptive metabolic reprogramming that occurs in pitaya plants under stress conditions, particularly in carbohydrate metabolism, which likely supports energy production and stress tolerance. The upregulation of genes involved in sucrose and starch metabolism indicates an enhanced capacity for energy storage and utilization, crucial for coping with abiotic stresses.

### 2.8. Photosynthesis-Related Genes Under Single and Combined Stresses

The single and combined stresses induced many DEGs related to photosynthesis. A total of 35 photosynthesis-associated DEGs were identified, three of which (PSB27-1, PSAN, PSAE-1) encode components of photosystem I (PSI) and exhibit slight upregulation in response to S and Cd stress treatments. In addition, CdD, CdSD, and CdDSM treatments also induced photosynthesis-related genes. A total of 35 genes encoding PSI and PS II were identified, with 11 showing upregulation (PSEA-1, PSAF, PSAK, PSAL, PSAN, PSB27-1, psbB, HCF136, PSBR, psbW) and 8 showing downregulation. Furthermore, numerous genes encoding redox chains showed significant up- or downregulation under different stress conditions. In addition, the expression of 411 genes, encoding components of chloroplasts, was significantly increased (Appendix A). The observations highlight the complex regulatory adjustments in the photosynthetic machinery of pitaya plants under abiotic stress, reflecting efforts to maintain photosynthetic efficiency and energy production. The upregulation of photosynthesis and chloroplast-related genes indicates an adaptive response aimed at optimizing light capture and electron transport during stress conditions.

### 2.9. Transcription Factors Responding to Single, Double, and Multifactorial Stresses

Important regulatory transcription factors (TFs) can either upregulate or downregulate genes that specifically bind to cis-acting regulatory elements in the promoters of target genes. In this study, a large number of TFs were found, including WRKY (64), NAC (43), MYB (80), bZIP (16), and bHLH (54). The WRKY TFs showed significantly altered expression in response to various treatments. A higher level of expression of certain genes was observed under single-factor stress treatments. The highest expression levels were reached by genes encoding WRKY70, WRKY41, WRKY40, WRKY44, and WRKY14 (*HU02G02628*, *HU11G01694*, *HU03G00514, HU04G00301, HU06G02430,* and *HU05G01710*), with 3.80-, 2.25-, 3.81-, 2.77-, 8.40-, and 6.63-fold upregulation, respectively. However, genes encoding WRKYs were downregulated under single, double, and multifactorial stresses, with fold changes ranging from −1.32 to −9.66. Thirty-five of the WRKY family genes showed lower expression levels in response to salinity (S) stress (Appendix A).

More than half of the genes encoding NACs were upregulated to varying degrees. Genes encoding NAC071, NAC092, NAC006, NAC090, NAC100, and NAC104 (*HU04G00486, HU01G02396, HU01G00420, HU07G02058, HU05G01179, HU03G00770*) reached the highest expression levels under single and combined stresses, with around 2.31-, 2.46-, 1.39-, 2.98-, 4.64-, and 4.3-fold upregulation, respectively. Interestingly, NAC100 (*HU05G01179*) was upregulated 4.64-fold under CdDM treatment conditions, but slightly decreased under Cd treatment, and then decreased from 4.84 to −5.38-fold in response to S stress, recording an increase of 2.71-fold under combined stresses. NAC6 (*HU01G00420*) exhibited a similar expression trend with slight upregulation. In contrast, a gene encoding NAC047 (Novel 8333) was downregulated in response to S and CdDSM, but its expression continued to increase thereafter (Appendix A).

More than 75% of the MYB genes exhibited varying degrees of upregulation in their expression levels under various multifactorial treatments. Among them, a sizable number of upregulated genes surfaced during Cd treatment. For example, MYB117 (*HU10G00373*) and MYB306 (*HU06G01167*) exhibited the highest degrees of upregulation at 5.69- and 4.64-fold, respectively. Additionally, under salinity (S) stress, a few genes showed a significant upregulation. With a 3.69-fold increase, MYB59 (*HU02G01484*) showed the highest level of upregulation among all upregulated genes after a single treatment. During multifactorial treatments, some genes tended to increase and then decrease. For instance, the gene encoding MYB61 (*HU08G01429*) showed a slight increase under S stress before its expression decreased 1.80-fold. After double and multifactorial analysis, only a small subset of gene expressions showed upregulation, while the majority remained relatively unchanged compared to control (Appendix A).

Several bHLH TFs were upregulated under single-, double-, and multifactorial stress treatments. The majority of the bHLH family’s genes were upregulated in response to S and Cd stress compared to other treatments. On the other hand, during the M-vs-Ck and CdS-vs-Ck comparisons, a significant number of genes were downregulated. A minor increase was observed among them, with an increase in regulation from 1.02 to 4.66-fold at Cd-vs-Ck. Still, 21 genes had been downregulated. One bHLH81-encoding gene (*HU02G02661*) was downregulated −4.83-fold (Appendix A). We undertake a thorough examination of the expression dynamics of pivotal transcription factor (TF) families in pitaya under diverse abiotic pressures. The analysis discerns the TFs and their distinct expression patterns, serving as crucial targets for further functional investigations and potential genetic modifications in order to enhance stress resilience in pitaya. The intricate regulatory mechanisms observed underscore the intricacy of the plant’s response to stress, indicating potential avenues via which future research can delve into the specific functions and interactions of these TFs in adapting to stress.

### 2.10. Melatonin Induces ROS-Scavenging and Circadian Clock Genes

Our research has shed light on how M affects gene expression in single and combined stresses. Our results showed that M regulates the expression of important genes related to stress responses, antioxidant defense, and circadian rhythms. Several genes related to antioxidant enzymes, such as glutathione peroxidase (GPX), catalase (CAT), and superoxide dismutase (SOD)—including *HU07G01008*; *HU02G03407*; *HU04G01183*; *HU05G00832*; *HU08G01598*; *HU01G01343*; *HU06G02305*; SPA1, *HU07G00071*; LHY, *HU11G01590*; NIN3; and *HU07G02013*—were upregulated. These are involved in regulating the antioxidant capacity of cells. In addition, the expression of genes associated with immune function, such as those encoding cytokines and chemokines involved in inflammation and the modulation of the stress response, was regulated by M. In addition, M influenced the expression of blue light photoreceptor activity genes such as CRY1 and CRY2. These are involved in the regulation of circadian regulation, which has implications for several physiological processes. Overall, these results demonstrate the diverse ways in which M influences gene expression to support and maintain cellular homeostasis.

### 2.11. Co-Expression Network Construction and Identification of WGCNA Modules

Gene clustering and module cutting in gene co-expression networks brought genes with comparable expression patterns together on the same branch. The expression patterns of 29,905 genes acquired from transcriptome data were assessed by WGCNA, and the unexpressed genes in more than half of the samples were filtered. Based on the similarity of expression patterns, 15 modules were found in total (Figure 4A). Heat map visualizations of the cluster analysis were used to assess the genes present in the modules (Figure 4B). In addition, the module–trait correlation relationships were built to determine significant correlation to single, double, and multifactorial stresses (Figure 4C). Six modules with attributes connected to nine treatments were present; the colors salmon, purple, black, pink, blue, and yellow were visible among them. The analysis of six selected modules’ expression patterns was performed, with the eigengenes’ expression pattern serving as a representation of the module’s overall gene expression profile (Figure 4).

GO and KEGG analyses revealed six annotated modules: salmon, purple, black, pink, blue, and yellow. The salmon module showed a significant enrichment of metabolic and catabolic processes, especially in the extracellular region, in anchored membrane components, and secondary metabolic processes. KEGG analysis showed THE primary enrichment of biosynthetic pathways. Under CdS stress, the purple module showed enrichment in terms pf amino sugar and nucleotide sugar metabolism, with the KEGG pathways highlighting the pentose phosphate pathways. The black module showed the enrichment of transcriptional regulator activities, including DNA-binding transcription factor and ubiquitin-protein transferase activities, as well as D recovery processes. KEGG analysis emphasized signal transduction as the primary form of enrichment (Figure 5; Appendix A).

In the GO analysis, the CdSD pink module was significantly enriched for nucleotide binding and stress responses, and the glutathione pathway was similarly highlighted in the KEGG analysis. Meanwhile, the blue module in CdSD was enriched in catabolic processes also showed significance in GO and KEGG analyses, highlighting the biosynthesis of additional secondary metabolites. In contrast, in the analysis of metabolic metabolism, the yellow module was highlighted. The salmon and blue modules were the main targets of the stress response terms in the GO analysis, suggesting that these modules play an important role in single- and combined-stress responses (Figure 6; Appendix A).

### 2.12. Identification of Hub TFs and Network Construction

In each module, several TFs were examined and most of them were related to WRKY, NAC, MYB, bHLH, and bZIP, although their distribution varied between modules. In the salmon module, numerous WRKY, MYB, NAC, and bHLH TFs were co-expressed (Figure 7A). Seventeen of these TFs were selected based on the hub gene correlation network and their high connectivity: WRKY (*HU02G02539, HU06G00411, HU01G00584*, and *HU10G01980*); NAC (*HU07G00854, HU02G00350, HU04G01575,* and *HU10G00380*); bZIP (*HU09G01935, HU09G01579,* and *HU10G00504*), MYB (*HU03G00868, HU06G01932,* and *HU06G00027*); and bHLH (*HU02G02661, HU04G01932*, and *HU01G00293*) (Figure 7B). Except for drought (D) stress, the experiment revealed that the expression of these genes was significantly upregulated in S, M, CdSD, and CdDSM (Figure 7B). Fifteen TFs, including WRKY, NAC, MYB, bHLH, and bZIP, were found to be hub genes in the blue module (Figure 7C). Higher connectivity was found between *HU02G01423*, *HU09G01037*, *HU06G02188, HU08G01835, HU04G02201, HU07G00854, HU01G00820*, and *HU10G00331* (Figure 7C). In S, M, CdSD, and CdDSM, the expression of all 15 TFs was higher (Figure 7C).

### 2.13. Validation of the DEGs by qRT-PCR

The results of the study show a strong correlation between RNA-seq and qRT-PCR data for several genes, indicating the reliable validation of the transcriptomic data. Specifically, five genes were analyzed, namely, *HU02062539*_WRKY, *HU01600584*_WRKY, *HU07060854*_NAC, *HU03060868*_MYB, and *HU01060293*_bHLH (Figure 8). For each gene, the results are shown in a scatter plot comparing the log2 fold values obtained from RNA-seq and qRT-PCR data. The correlation coefficients (r) and regression equations displayed on the plots demonstrate a high level of consistency between the two methods. For *HU02062539*_WRKY, the plot shows a significant correlation (r = 0.9176), indicating that the gene expression changes detected by RNA-seq data are reliable and consistent with qRT-PCR validation.

## 3. Discussion

### 3.1. The Effect of Stress on Plant Performance Is Enhanced by Melatonin

Cd exposure exhibited the most pronounced detrimental effects on the growth of dragon fruit seedlings compared to other single stressors. Research on *Eleocarpus glabripetalus* seedlings, which bear some similarities to pitaya, demonstrated that the negative impacts of Cd can be mitigated through the addition of nitrogen. This study revealed that nitrogen enhances root system development and overall biomass, thereby ameliorating the adverse effects of cadmium on plant growth [20,21]. Additionally, studies on *Phragmites australis* indicated that Cd impedes germination and early seedling growth in a dose-dependent manner, suggesting that there may be similar effects on pitaya under analogous conditions [22].

When combined with S or D, Cd stresses further inhibited growth, particularly in the Cd and S combination. The application of M, however, attenuated the impacts of these multifactorial stresses on dragon fruit seedlings, except in the simultaneous presence of Cd, S, and D. Notable reductions in morphological parameters such as cladode length, diameter, and the size of shoots and roots underscore the severity of these stresses (Figure 1). Despite increased Cd accumulation under combined stresses, the application of M countered these negative effects and fostered growth. These findings align with previous research, indicating the potent inhibitory influence of Cd on plant growth. A study on *Wolffia arrhiza*, a small aquatic plant, showed that pretreatment with M significantly boosted levels of photosynthetic pigments, proteins, and monosaccharides, diminishing the toxic effects of Cd. This effect was particularly pronounced in plants concurrently treated with 25 µM of M and Cd, highlighting M’s potential to mitigate the adverse effects of cadmium on plant growth [23]. Another investigation into the combined effects of silicon and M on maize under cadmium stress conditions demonstrated that this synergy markedly improved growth parameters like plant height and stem diameter, and also reduced cadmium accumulation, further underscoring its protective benefits against cadmium toxicity [24]. Additionally, the M treatments have been shown to alleviate the adverse impacts of Cd, D, and S stresses by bolstering the plant’s inherent protective mechanisms [25]. Understanding the molecular mechanisms that govern plant responses to combined stresses is crucial. This is exemplified by the identification of 141 common DEGs, primarily associated with heavy metal tolerance, across all the stress conditions analyzed. In tomato plants subjected to Cd stresses, significant transcriptional alterations were observed, including the expression of 1123 DEGs in roots and 159 in shoots. Expression was enriched in pathways that are crucial for countering oxidative stress, such as glutathione and sulfur metabolism. These findings underscore the complex regulatory networks that plants employ in response to Cd stress [26]. In *Zoysia japonica*, a turfgrass with potential phytoremediation capabilities, transcriptome analysis post-Cd exposure revealed the presence of between 5321 and 6526 DEGs over time, indicating robust gene regulation in response to Cd [27].

### 3.2. The Common and Unique Transcriptome Response and Cross-Talk of Single and Multiple Stresses

The transcriptome analysis of white pitaya under diverse stress conditions unveiled both distinct and overlapping DEGs in response to single-, double-, and multifactorial stress treatments. When subjected to single-stress conditions such as D, M, and Cd, we identified 3805 unique and 141 common DEGs. Notably, *HU04G00348*, *HU02G00171*, and *HU07G00210* were significantly upregulated under drought stress, underscoring their roles in protein degradation, osmotic regulation, and stress signaling, which are critical for maintaining cellular homeostasis and adapting to water scarcity. This is consistent with previous studies that have highlighted the importance of these processes in drought stress responses in other plant species [28,29].

Meanwhile, *HU04G00192* and *HU05G01710* exhibited unique expression patterns under metal stress, emphasizing their involvement in melatonin-modulated regulatory mechanisms. *HU02G03171* was exclusively responsive to S stress, while *HU02G02576* was specifically responsive to cadmium exposure. These findings align with prior research indicating that melatonin plays a pivotal role in modulating stress responses and enhancing plant tolerance to abiotic stresses [30].

In double-stress treatments (CdS and CdD), we observed both unique and common DEGs. *HU09G01164* was uniquely upregulated under CdS treatment, highlighting its specific role in combined responses to metal and salinity stress. Interestingly, *HU09G00269* and *HU10G00959* were commonly regulated under both CdS and CdD, suggesting the existence of shared pathways that integrate responses to metal and abiotic stress and indicating a convergent regulatory mechanism where certain genes are pivotal to mediating responses to multiple stress factors. Similar integrative responses have been reported in Arabidopsis, where common signaling pathways are activated under combined-stress conditions [31].

Multifactorial treatments further complicated the expression patterns. For instance, in treatments like CdS-vs-CdDSM and CdD-vs-CdDM, *HU06G00171* exhibited differential expression, indicating the existence of a complex regulatory mechanism when multiple stresses are combined. *HU05G01506* was differentially expressed in CdS-vs-CdSM, illustrating stress responses’ multifaceted nature. Additionally, *HU10G03180* was identified as a common DEG across all multifactorial treatments, underscoring its crucial role in the integrated stress response. These results are in line with findings from other plant species, where complex regulatory networks are activated to cope with multifactorial stresses [32].

Our findings underscore the intricate crosstalk between different stress responses in white pitaya. The genes *HU04G00348, HU02G00171*, and *HU07G00210* exhibit stress-specific expressions, while others, such as *HU09G00269* and *HU10G00959*, participate in a common regulatory network. This highlights the sophisticated balance plants maintain to adapt to varying and combined environmental challenges. Prior studies have also noted the importance of such regulatory networks in ensuring plant survival under diverse environmental conditions [33,34].

This underscores the sophisticated balance plants maintain to adapt to varying and combined environmental challenges.

### 3.3. Enhanced Antioxidant Activities in Pitaya Under Combined-Stress Conditions

This study elucidates significant biochemical changes in plants subjected to single- and combined-stress treatments, revealing that combined stresses generally induce stronger biochemical responses than single stresses, reflecting a synergistic effect on the plant’s stress physiology (Figure 3). The observed increases in H_2_O_2_ content across all stress treatments indicate an enhanced oxidative stress response. The M treatment exhibited the highest increase, with a 175% rise compared to the control. Single stresses (Cd, S, and D) showed increases of 50%, 100%, and 125%, respectively, while combined stresses (CdS and CdSD) resulted in a 150% increase. These findings align with previous studies suggesting that combined-stress treatments exacerbate oxidative stress more than individual stresses, consistent with the enhanced ROS production seen under stress conditions [35]. APX activity, a crucial antioxidant enzyme, was notably elevated in the S and D treatments, with increases of 185% and 200%, respectively, compared to the control. Combined stresses, such as CdSD, induced an 85% increase. This elevated APX activity under stress conditions is consistent with its role in detoxifying H_2_O_2_ and mitigating oxidative damage, as previously reported by Mittler (2017) [36]. POD activity followed a similar pattern to APX, with the highest levels observed in the S and D treatments, showing increases of 150% and 160%, respectively, compared to the control. Combined stresses (CdS and CdSD) showed increased POD activity, with rises of 120% and 110%, respectively. These results suggest the existence of an enhanced antioxidative defense mechanism under combined-stress conditions, corroborating the findings of [37]. CAT activity was highest in control plants. Single stresses (S and D) significantly reduced CAT activity by 80% and 60%, respectively. However, combined stresses (CdS and CdSD) maintained or slightly enhanced CAT activity by 20% and 15%, respectively. This indicates that while single stresses severely impact CAT activity, combined stresses may induce compensatory mechanisms to maintain CAT levels, as observed in studies by Gill et al. (2010) [38]. SOD activity was elevated in the S and D treatments by 57% and 71%, respectively, compared to the control, and was even higher under combined stresses like CdSD, showing an 86% increase. SOD, being the first line of defense against ROS, plays a crucial role in mitigating oxidative damage, as evidenced by its increased activity under stress conditions [39]. Proline content, a known osmoprotectant, was significantly higher in the S treatment, showing a 175% increase compared to the control. Combined stresses (CdS, CdSD, and CdSM) further boosted proline levels by 138%, 150%, and 125%, respectively. Proline accumulation under stress conditions helps to stabilize proteins and membranes, suggesting an enhanced protective response under combined stresses, in line with the findings of [40]. The findings of this study indicate that combined-stress treatments exacerbate oxidative stress and activate more robust antioxidative defense mechanisms compared to single stresses.

### 3.4. Transcription Factors Involved in the Response to Single and Combined Stress

In our investigation, several TFs belonging to the WRKY, MYB, NAC, and bHLH families were discovered to be co-expressed within the salmon module, indicating a complex interplay within the gene network during stress conditions. Previous research has elucidated the intricate interactions of these transcription factors under stress conditions in plants. Notably, the co-expression of WRKY, MYB, NAC, and bHLH transcription factors within the salmon module has been associated with carotenoid synthesis in plants. These TFs were identified through WGCNA and demonstrated a significant correlation with carotenoid expression, emphasizing their critical role in stress responses and metabolic pathways [41]. Their structural features, such as the WRKY domain and zinc finger-like motifs, facilitate specific DNA binding that regulates genes responsive to stress [42]. In potatoes, the analyses of co-expression and differential expression highlighted how different transcription factor families (WRKY, MYB, bHLH, and bZIP) respond under abiotic stresses like heat, salinity, and drought, with specific TFs showing high elvels of expression under certain conditions [43]. Bread wheat analysis, involving genome-wide analysis and miRNA targeting under stress conditions, identified numerous TFs across these families in response to abiotic stresses [44]. Research on *Populus* species under multiple abiotic stresses demonstrated significant upregulation of TFs from the WRKY, MYB, and bHLH families, enhancing stress tolerance in transgenic lines [45]. Studies on Arabidopsis have shown the role of WRKY TF in modulating S stress responses, particularly highlighting the antagonistic functions of WRKY8 and its interacting partners in tolerating salinity stress [46].

In this study, eight WRKY genes, including WRKY75, WRKY28, and WRKY27, exhibited high connectivity within the salmon module, suggesting their critical roles in responding to both single- and multifactorial stress conditions. A study on Arabidopsis thaliana demonstrated that WRKY25, WRKY26, and WRKY33 coordinate to induce thermotolerance by regulating the transcriptional changes of heat-inducible genes in response to elevated temperatures [47]. Further research highlighted the role of the mitochondrial protease FtSH4 contained in Arabidopsis in regulating WRKY-dependent salicylic acid accumulation and signaling, thus influencing leaf senescence and defense mechanisms [48]. Complex regulatory networks have been documented that involve WRKY TFs and other TF types, like bZIPs, in the context of unfolded protein responses and immune regulation in Arabidopsis, thus enhancing our understanding of how WRKYs affect broader physiological and defense responses [49]. For instance, PagWRKY75 was found to be downregulated during the initial stages of osmotic and salt stress, and transgenic poplar lines overexpressing PagWRKY75 exhibited increased sensitivity to these stresses [50].

Fifteen TFs from the blue module, belonging to the WRKY, NAC, MYB, bHLH, and bZIP families, were recognized as hub genes, showing enhanced expression in response to stress conditions (Figure 7). Genome-wide analysis in tomato highlighted the roles of WRKY and bHLH TFs in response to Cd stresses, providing insights into their regulatory functions in stress response pathways. This potentially includes combined stresses like salt and drought [51]. Additionally, a comprehensive review of the roles of various TFs in plants under multiple abiotic stresses discussed how these TFs interact with cellular components to modulate stress responses, essential for developing stress-tolerant plant varieties [52]. A detailed study on wheat TFs explored their diverse expression patterns under abiotic stresses, highlighting the functional importance of families such as WRKY, MYB, and bZIP in managing combined-stress conditions [53].

### 3.5. Phytohormone Signals Under Single and Combined Stresses

Under Cd stress, the IAA4 gene was significantly upregulated, exhibiting a 7.24-fold increase in expression. Conversely, the ABP19B gene was notably downregulated 11.98-fold under the same conditions. This research also highlighted the association of abscisic acid (ABA) synthesis with the downregulation of the ABA 8 and hydroxylase CYP707A2a genes during Cd exposure. These significant changes are accompanied by increases in plant ABA levels and are pivotal in both single- and multifactorial stress responses. In the ABA synthesis pathway, indole-3-acetic acid (IAA) is identified as a crucial rate-limiting enzyme [54]. ABA serves as an endogenous signal, mediating abiotic stress responses in plants, and AtIAA4 is involved in ABA biosynthesis during seed development [55].

Furthermore, the study observed varied responses in ethylene biosynthesis genes under stress conditions. One ACC synthase (ACS) gene was upregulated, while the expression of two ACC oxidase (ACO) genes was also increased, yet three ACS genes were downregulated. Currently, it remains unclear whether D stress specifically leads to an increase or decrease in ethylene levels. Ethylene is known to play a significant role in various developmental processes and stress responses in plants [56]. Notably, plants under D stress, such as pineapples, produce significantly less ethylene in leaf and stem tissues compared to control plants [57]. These divergent findings suggest species-specific responses to stress. Generally, changes in ACC biosynthesis during D stress can impact ethylene production. The ethylene response factors (ERFs), which are downstream elements of the ethylene signaling pathway, are critical in plant responses to abiotic stresses [58].

Additionally, this study found that two brassinosteroid LRR receptor kinases (BRL) genes showed increased expression in response to single and double stresses, suggesting a potential defensive role against abiotic stressors. BRs and ABA are known to mediate plant responses to abiotic stress, with recent studies indicating that BR plays a role in modulating responses to single, double, and multifactorial stresses [59]. The interplay between BR and ABA signaling pathways demonstrates an antagonistic effect, where BRs are thought to mitigate responses to stress while ABA enhances them [60]. Research suggests that Arabidopsis can develop drought resistance without normal growth being comprised via the overexpressing of BRL2, a vascular-rich member of the BR family [61]. Such findings underscore the importance of studying hormonal interactions in model plants to better understand their roles in stress tolerance, particularly in species like pitaya. This research not only sheds light on the hormonal dynamics during stress conditions but also lays a foundational understanding of the interactions among hormones in single- and multifactorial stress responses in plants.

### 3.6. Phytohormone Signals Under Single and Combined Stresses

This study investigated the transcriptional responses of white pitaya (*S. undatus* L.) under various stress conditions, revealing 3805 unique and 141 common DEGs across single-stress treatments involving D, M, S, and Cd exposure. Key genes, such as *HU04G00348, HU02G00171*, and *HU07G00210*, were significantly upregulated under D stress, indicating roles in protein degradation, osmotic regulation, and stress signaling, crucial for maintaining cellular homeostasis during water scarcity. Unique DEGs were identified for M (*HU04G00192, HU05G01710*), S (*HU02G03111*), and Cd exposure (*HU02G02576*), highlighting specific regulatory mechanisms. These findings suggest that white pitaya utilizes a complex array of genes to manage different stress conditions. The upregulation of *HU04G00348, HU02G00171*, and *HU07G00210* under drought stress conditions underscores their importance in maintaining cellular homeostasis and adapting to water scarcity. The unique expression patterns observed for melatonin, salinity, and cadmium stresses point to specialized pathways, activated in response to these stressors. Previous studies have documented the role of specific genes in abiotic stress responses in other plant species [62]. Our findings align with this body of work, but also extend it by identifying new genes involved in white pitaya’s response to combined-stress conditions. For instance, the unique upregulation of *HU09G01164* under combined cadmium and salinity stress suggests a novel role in managing metal and salinity stress. This research is significant for agricultural practices and crop improvement strategies, offering practical benefits for developing stress-resilient white pitaya cultivars.

### 3.7. Signaling Pathway Mediates Single- and Combined-Stress Responses

In this study, we analyzed gene expression under various stress conditions in *S. undatus* L. Specifically, six genes were upregulated under S stress conditions, two during M treatment, three under drought D stress conditions, seven during Cd stress, and two during combined Cd and D stress, and four were downregulated under S stress. The application of M treatment in combination with the stresses Cd, S, and D revealed the involvement of CaM/CMLs. Changes were particularly noted in the expression of genes such as CML35, CML39, CML48, CML42, CML44, CML41, CML18, and CML13. Interestingly, the genes encoding CML13 and CML49 were downregulated.

Our findings are consistent with prior research, indicating that CDPKs/CPKs are critical in several stress signaling pathways. Notably, D and S stress downregulated genes such as AtCML8, AtCML13, AtCML18, and AtCML25 [63]. In addition, it has been reported that under cold, D, and S stress conditions, the *ShCML44* gene shows elevated expression in *Salvia hispanica*. The overexpression of the *HuERF1* gene in pitaya plants has been shown to have beneficial effects. In particular, it helps to improve antioxidant activity and improves salt tolerance by regulating reactive oxygen species (ROS) levels under exposure to salt stress. This implies that different types of stress may have similar response mechanisms [64].

Transcriptome analysis under drought conditions revealed significant changes in pitaya hormone-mediated signaling pathways. Furthermore, this study identified complicated interactions between the auxin and ethylene signaling pathways [65]. Some research highlights the need for complex physiological and biochemical mechanisms to ensure pitaya drought tolerance by identifying numerous DEGs that contribute to D responses [66]. Furthermore, the results reveal a network of interconnected stress responses within the plant, as Cd exposure affects the expression of genes involved in other stress-responsive pathways [67].

Our study also showed that six genes encoding putative calcium-dependent protein kinases were upregulated under single, double, and multifactorial stresses. Four of these genes exhibited high expression levels during S, D, Cd stress, as well as combined treatment with Cd, S, and D with M. These findings imply the significant activation of the AtCBP and ALA signaling pathways under both single- and multifactorial treatments. Interestingly, three genes encoding CBP were found to be significantly downregulated under S, D, and combined Cd and S stress conditions with M treatment, suggesting a role for the CBP signaling pathway in managing abiotic reactions in *S. undatus* L.

The above results underscore the pivotal role played by the calcium-mediated signaling pathway in the response of *S. undatus* L. to both single and multifactorial stressors. Abiotic stress tolerance in Arabidopsis is conferred by the overexpression of AtCBP and ALA, as demonstrated in a study [68]. AtCBP1, on the other hand, plays a role as a negative regulator of the osmotic response, emphasizing the importance of these pathways. Research has shown that distinct calcium-dependent protein kinases are crucial for enhancing tolerance to S and D stress [69]. Moreover, extensive transcriptome analysis has revealed that under stress conditions, numerous genes are either upregulated or downregulated, contributing to the plant’s overall stress management strategy [70].

### 3.8. Metabolism Response to Single and Combined Stresses

Our study shows that *S. undatus* L. is a species that responds strongly to various stimuli by adjusting its metabolism. We noticed an increase in the production of osmoprotective substances is crucial to improving the plant’s resistance to harsh environmental conditions. Proline, an amino acid, plays a crucial role in osmoprotection and helps to promote growth when the plant is under osmotic stress. This amino acid is converted into different molecules through different enzyme pathways during this process [71]. To endure such conditions, Chlamydomonas significantly elevates the synthesis of intricate lipids and alters its fatty acid profiles [72]. Furthermore, higher levels of CO_2_, which can act as a form of environmental stress in CAM plants, like *Hylocereus undatus* and *Selenicereus megalanthus*, result in enhanced carbon assimilation and development, indicating an increase in sugar and potentially starch production in roder to adapt to environmental fluctuations [73].

Additionally, genes related to lipid biosynthesis, particularly those of the long-chain acyl-CoA synthetases (LACS) family, showed varied expression under stress conditions. LAC3 and LAC12, for example, were induced under combined Cd and D stress and magnesium (M) treatment, while several LACS genes were downregulated under S and Cd stress. This highlights the crucial role of lipid metabolism in the adaptation of *S. undatus* L. to stressful conditions, aligning with findings from *Chromochloris zofingiensis* that provide insights into the functions of LACS in lipid metabolism under stress [74].

Secondary metabolism also plays a significant role in the adaptation of plants to their environment under both abiotic and biotic stresses. The phenylpropane pathway, which is particularly responsive to various abiotic stresses, including heatwaves and drought, is pivotal in this regard. In white grapes (*Vitis vinifera* L.), the activation of this pathway in response to drought stress involves the significant regulation of genes encoding enzymes like coumaric acid-CoA ligase (4CL) and phenylalanine ammonia-lyase, which are crucial in the biosynthesis of flavonoids and other phenylpropanoids [75].

Our study further revealed that single and multifactorial stresses induced the expression of genes encoding enzymes involved in flavonoid biosynthesis, such as chalcone isomerase (CHS), anthocyanin synthase, and flavonoid synthase (FLS). These enzymes are integral to the synthesis of flavonoids, which play key roles in UV protection and antioxidative defense mechanisms, potentially enhancing stress tolerance under conditions like Cd and drought stress. In wheat, the involvement of CHS in flavonoid biosynthesis links directly to plant adaptive responses to environmental stress, with varied transcriptional responses to salinity stress across seen different homologous genes [76].

Furthermore, studies have shown that anthocyanin synthase contributes significantly to the biosynthesis of anthocyanins, pigments that protect plants against various stresses by scavenging free radicals and enhancing stress tolerance [77]. The regulation of flavonoid synthesis through enzymes like FLS is critical under stress conditions, as evidenced by research indicating that BRs signaling modulates the expression of key enzymes in flavonoid biosynthesis, illustrating a tradeoff between growth and stress responses, including UV-B protection [78].

These complex interactions highlight the sophisticated mechanisms plants employ to manage stress through metabolic adjustments and the transcriptional regulation of key pathways, underscoring the potential for genetic and biotechnological interventions to enhance stress tolerance in crops.

## 4. Materials and Methods

### 4.1. Plant Materials and Stress Treatments and Melatonin Application

The seeds of the red peel white pulp pitaya (*Selenicereus undatus* L.) were sown in a controlled environment at the Sanya Nanfan-Research Institute, located at Yazhou Bay Seed Science & Technology City, Hainan University, Haikou, China. These seeds were sourced from Hainan-Shengda Modern Agriculture Development Company, Qionghai, Hainan, China. For the initial growth phase, pitaya seeds were placed on filter paper. After six days of germination, uniform seedlings were transferred to hydroponic pots enriched with Hoagland solution [79]. The seedlings were cultivated in a controlled growth chamber with a 16 h light/8 h photoperiod, a constant temperature of 24 °C, and a relative humidity of 70%. After 30 days, the plants underwent various stress treatments and melatonin (M) applications containing the following components: 200 mM NaCl, 20% PEG-600, and 1.5 mM Cd. These chemicals were applied to plants growing in hydroponic solution pots. These chemicals were sourced from Sinopharm Chemical Reagent Co., Ltd., Shanghai, China. Samples were collected seven days after the stress application. Additionally, M was applied at a concentration of 0.099 μM to plants under natural growing conditions and also two days before exposure to combined stresses of S, D, and Cd. This combined treatments were termed CdSM, CdDM, and CdDSM for the respective stress combinations.

### 4.2. Phenotyping and Quantification of Cd Content

Various morphological parameters were measured to assess the impact of different treatments on pitaya seedlings. These parameters included CL, CD, SL, SW, RL, and RW. The Cd content in pitaya shoots and roots was quantified using the established method with a graphite digestion device (GY-25), developed by Zhejiang Tuojie Instrument Co., Ltd. (Shaoxing, China).

### 4.3. RNA Extraction, cDNA Library Construction, and Sequencing

Total RNA was extracted from 0.5 g of pitaya cladodes using the RNAprep Pure Total RNA Extraction Kit (TIANGEN, Beijing China). The samples from the emerging stems (cladodes) and leaves were obtained from 11 treatments. The RNA was extracted from all the collected samples and the quality and integrity of RNA were assessed using the Agilent 2100 bioanalyzer (Agilent technologies, Santa Clara, CA, USA). To remove genomic DNA contamination, 1 μg of RNA from each sample was treated with DNase I according to the kit instructions and then used to generate cDNA with the QuantiTect Reverse Transcription Kit (Qiagen, Shanghai, China). The RNA library was prepared using the NEBNext Ultra-RNA Library Prep Kit for Illumina (NEB, Ipswich, MA, USA). RNA sequencing was performed using the Illumina NovaSeq 6000 (NEB, Ipswich, MA, USA) with the S4 kit components from Beijing Novogene Technology Co., Ltd., Beijing, China. RNA-seq data were deposited under bioproject accession number PRJNA1121504.

### 4.4. De Novo Assembly, and Annotation

The raw sequencing data, originally given in a FASTQ format, were processed using NGSToolkits (version 2.3.3) with default parameters. To ensure data reliability and cleanliness, sequences containing adapters, poly-Ns, or low-quality reads were filtered out of the raw data. De novo assembly of high-quality data was performed using the Trinity method as described by Song et al. (2019) [80], which led to the generation of unigenes.

### 4.5. Identification of Differential Gene Expression (DEGs)

After filtering out low-quality reads, Illumina sequencing reads were aligned to the *S. undatus* L. reference genome (available at http://www.pitayagenomic.com/blast.php (accessed on 25 March 2023) using HISAT with default parameters [81]. The resulting BAM files containing uniquely mapped reads served as inputs to HTSeq. Gene expression levels were quantified using the FPKM (fragments per kilobase of transcript per million mapped reads) method [82]. To determine normalized expression levels, a log2(FPKM + 1) transformation was applied, and the Pearson correlation coefficient between biological replicates was calculated. DEGs were identified using the Cuffdiff tool based on the methods described by Konishi (2016) [83], with a log2 fold change (FC) threshold and false discovery rate (FDR) ≤ 0.05.

### 4.6. Enrichment Analysis of DEGs

GOseq was used to identify significant GO terms (*p* < 0.02) for the gene DEGs in the significant modules, following the methodology described by Ye et al. (2019) [84]. The cluster profiler package in R 4.2.3 was used to perform KEGG enrichment analysis on the DEGs with a significance level of *p* < 0.02.

### 4.7. Identification of Hub Genes and Sub-Networks Associated with Treatments

The hub genes with strong connections with the top 25% of genes with the highest centrality values were determined using the CytoNCA plugin of Cytoscape v3.7.2. Centrality measures, such as between centrality (BC), closeness centrality (CC), and degree centrality (DC), were utilized. The STRING database was used to construct protein–protein interaction networks, with the confidence score threshold set to >0.4. Hub genes were identified among the DEGs in response to various treatments. Cytoscape v3.7.2 was used to identify sub-modules within the network of DEGs [85]. Co-expression network analysis was carried out using the weighted gene co-expression network analysis (WGCNA) (v1.47) packages in R 4.2.3 [86].

### 4.8. Enzymatic Assay

To assess the enzymatic activity, 0.5 g plant samples were ground in 5 mL of 50 mM PBS with a pH of 7.8 [87]. Samples were then centrifuged at 10,000 RPM for 10 min at 4 °C. The antioxidant enzyme activities, H_2_O_2_ content, and osmoregulatory substances were determined using kits provided by Suzhou Michy Biomedical Technology Co., Ltd. (Suzhou, China). The activity of superoxide dismutase (SOD, M0102A) was measured using the WST-8 method [88], the activity of peroxidase (POD, M0105A) was assessed using the guaiacol method [89], the activity of catalase (CAT, M0103A) was measured using the UV absorption method [90], and the activity of ascorbic peroxidase (APX, M0403A) was also measured using the UV absorption method [91]. The H_2_O_2_ content was determined via the titanium sulfate method [92]. Additionally, the proline content was measured using the ninhydrin method [93], and the protein content was measured using the BCA method [94]. All assays were performed according to the manufacturer’s instructions, with the values of different antioxidants, including SOD, CAT, APX, PRO, POD, and H_2_O_2_, computed at wavelengths of 450, 240, 340, 412, 520, 470, 450, and 340 nm, respectively.

### 4.9. RNA Extraction and qRT-PCR

Total RNA was extracted from plant material using RNAprep Pure Plant Kit (TIANGEN, Beijing, China) according to the manufacturer’s instructions. RNA extraction was performed using leaves or cladodes of each treatment group, including single and combined groups. Three samples from each replicate were pooled to produce an RNA extract following a previously described approach [95]. Data analysis was performed using the 2^−ΔΔCT^ method [96], which accurately quantified gene expression by comparing the expression of the target gene with that of the reference gene. The Actin gene (*HU07G00802.1*) was utilized as an internal control. This gene has been validated as a stable internal control in previous studies [97]. The primers used in this study are given in Appendix A.

### 4.10. Statistical Analysis

Using SPSS 23.0 (SPSS Incorporation, Chicago, IL, USA), statistical analysis was performed with a significance level of ≤0.05 via ANOVA and Duncan’s test. The mean ± standard deviation of the three biological replicates was used to describe the data.

## 5. Conclusions

In this study, transcriptome analysis provides a vital resource for understanding pitaya’s stress-responsive genes. It reveals various cellular responses to single, double, and multifactorial stresses, highlighting the plant’s need for specific or antagonistic responses to elevated heavy metal levels, whether applied alone or in combination with other abiotic stresses. Notably, Cd exposure inhibited pitaya growth, affecting stem and cladode development, and its adverse effects were intensified when combined with D and S stresses. These combined-stress factors synergistically reinforced each other’s impacts, particularly with Cd, D, and S together. Interestingly, Cd combined with M had a less severe effect on growth compared to S and D combined. However, the combined stress of Cd, S, and D severely compromised pitaya, significantly increasing Cd uptake. Studies have shown that salinity can enhance Cd uptake due to increased transpiration and stomatal opening, which facilitates greater Cd absorption. This study successfully mapped gene regulatory networks and identified TFs associated with abiotic stress tolerance, enhancing our understanding of *S. undatus* L.’s response to various stresses. Transcriptome data indicate that signaling receptors, including ion channel proteins, play a key role in sensing and transmitting signals through calcium (Ca^2+^), ROS, and phytohormone pathways in response to abiotic stress. These signals are relayed by protein phosphorylases and kinases, activating TFs that are critical for cellular homeostasis and triggering downstream gene transcription responsive to S, D, and Cd stresses, including lipid, carbohydrate, and secondary metabolism. WGCNA identified hub TFs related to single-, double-, and multifactorial stress responses in pitaya. Furthermore, this study identifies the significant role of M in mitigating stress effects in pitaya. We found that M regulates gene expression and enhances stress tolerance, highlighting its potential as a key factor in improving resilience to abiotic stresses. Our findings suggest that further exploration of these physiological and genetics mechanisms, and their interactions with other stress-related pathways, could lead to effective strategies for improving the resistance to environmental stressors in different plants.

## Figures and Tables

**Figure 1 ijms-25-08901-f001:**
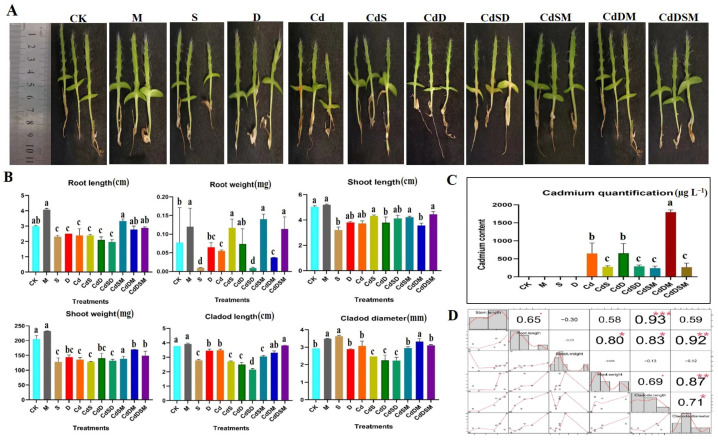
The morpho−physiological response of pitaya in under single-, double-, and multifactorial stress combinations: (**A**) The effect of single and combined stresses on the growth of pitaya plants. (**B**) Phenotypic measurements, including root length (RL), root weight (RW), shoot length (SL), shoot weight (SW), cladode length (CL), and cladode diameter (CD). The treatments were compared using one-way ANOVA at a 5% (*p* ≤ 0.05) probability level, and means were compared using the Duncan test. The different letters display statistically significant differences among cultivars; similar letters depict statistically non-significant values, while bars show standard errors. (**C**) Cadmium quantification across treatments, with error bars representing standard deviation. (**D**) A correlation matrix showing significant relationships among RL (cm), RW (g), SL (cm), SW (g), CL (cm), and CD (mm). Significant and highly significant correlations are represented by * (*p* < 0.05), ** (*p* < 0.01), *** (*p* < 0.001). Abbreviations: control (Ck); melatonin (M); salinity (S); drought (D); cadmium (Cd); cadmium + salinity (CdS); cadmium + drought (CdD); cadmium + salinity + drought (CdSD); cadmium + salinity + melatonin (CdSM); cadmium + drought + melatonin (CdDM); cadmium + drought + salinity + melatonin (CdDSM).

**Figure 2 ijms-25-08901-f002:**
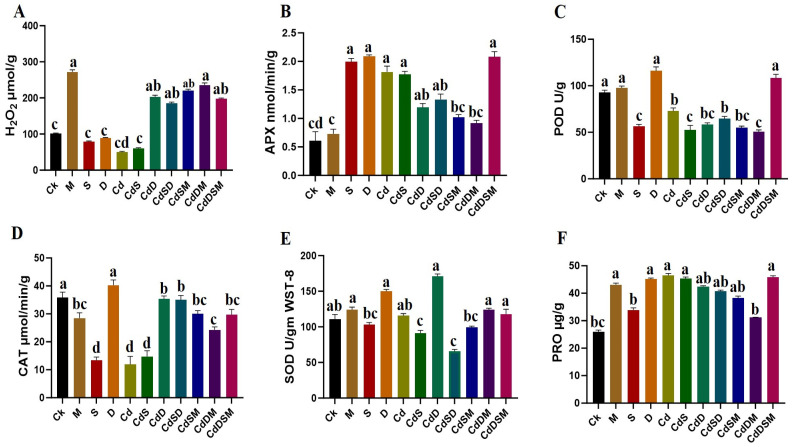
Physiological response of pitaya in response to single, double, and multifactorial stresses at seedling stage. (**A**) H_2_O_2_ concentration (µmol/g); (**B**) ascorbate peroxidase (APX) activity (mmol/min/g); (**C**) peroxidase (POD) activity (U/g); (**D**) catalase (CAT) activity (µmol/min/g); (**E**) superoxide dismutase (SOD) activity (U/g WST-8); and (**F**) proline (PRO) content (µg/g). Each bar represents mean ± standard deviation (*n* = 3), and different letters above bars indicate significant differences among treatments at *p* ≤ 0.01. Abbreviations: control (Ck); melatonin (M); salinity (S); drought (D); cadmium (Cd); cadmium + salinity (CdS); cadmium + drought (CdD); cadmium + salinity + drought (CdSD); cadmium + salinity + melatonin (CdSM); cadmium + drought + melatonin (CdDM); cadmium + drought + salinity + melatonin (CdDSM).

**Figure 3 ijms-25-08901-f003:**
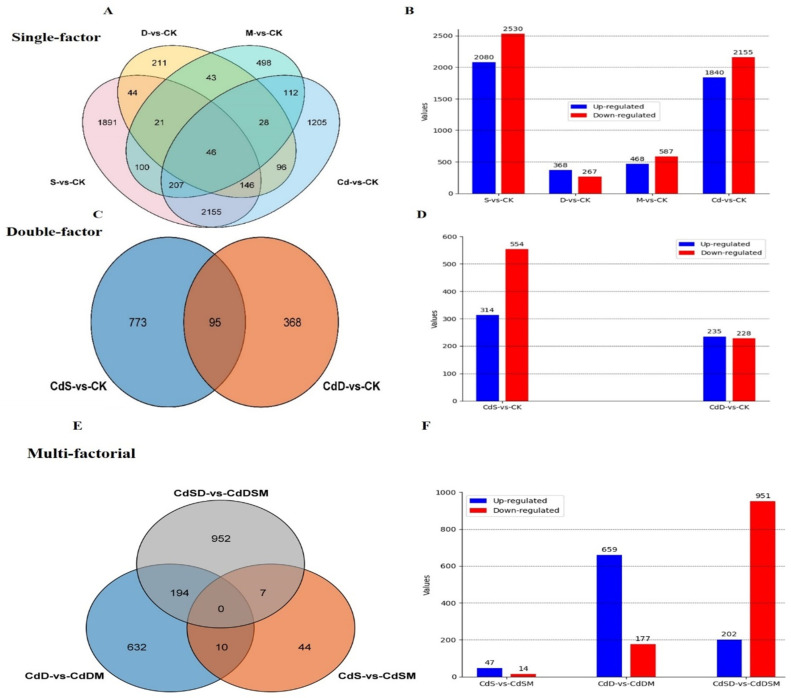
Identification of total and common differentially expressed genes (DEGs) in response to single-, double-, and multifactorial stress treatments (**A**,**B**). Common upregulated and downregulated DEGs for single-stress treatments: drought vs. control (D-vs-Ck), melatonin vs. control (M-vs-Ck), salinity vs. control (S-vs-Ck), cadmium vs. control (Cd-vs-Ck); (**C**,**D**) DEGs for double-stress treatments: cadmium + salinity vs. control (CdS-vs-Ck), cadmium + drought vs. control (CdD-vs-Ck); (**E**,**F**) DEGs for multifactorial treatments: cadmium + salinity + drought vs. cadmium + drought + salinity + melatonin (CdSD-vs-CdDSM), cadmium + drought vs. cadmium + drought + melatonin (CdD-vs-CdDM), cadmium + salinity vs. cadmium + salinity + melatonin (CdS-vs-CdSM).

**Figure 4 ijms-25-08901-f004:**
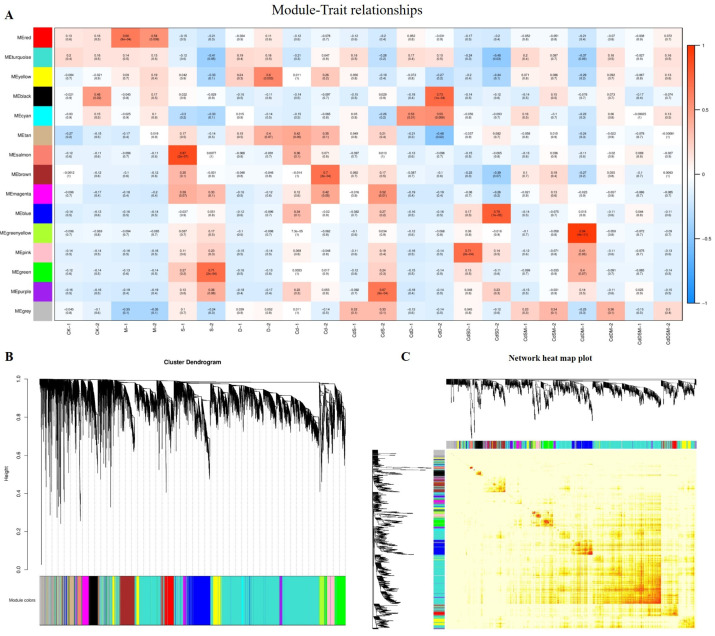
Weighted gene co-expression network analysis (WGCNA). (**A**) Module−trait relationship: each column represents different processing conditions. Each row corresponds to characteristic gene of module. Correlation between two is indicated in each cell by Pearson correlation coefficient and *p*-value in parentheses. Cell color ranges from red (indicating high positive correlation) to green (indicating high negative correlation). Number of genes contained in each module is displayed in parentheses on left side. (**B**) Module level clustering diagram. (**C**) Network heatmap plot.

**Figure 5 ijms-25-08901-f005:**
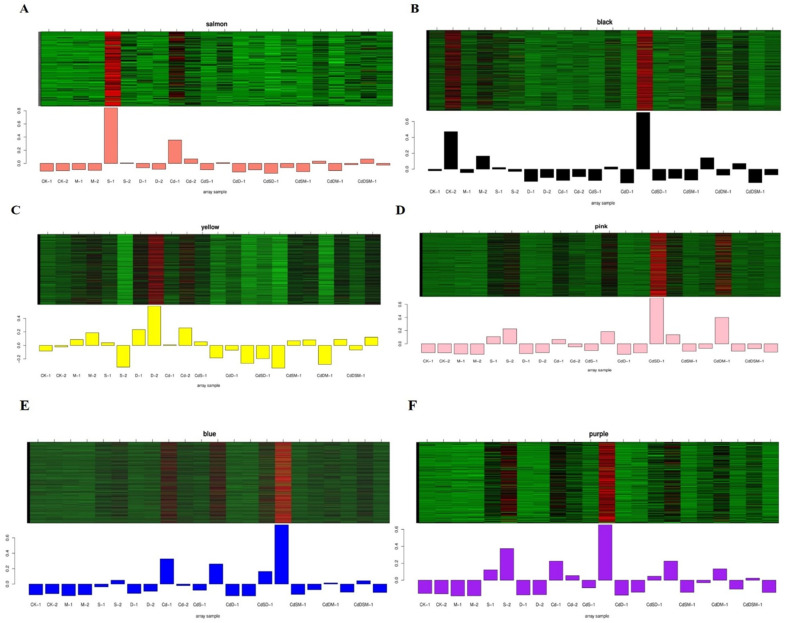
The expression pattern of the co-expressed genes in the representative module. (**A**) Salmon module; (**B**) black module; (**C**) yellow module; (**D**) pink module; (**E**) blue module; (**F**) purple module. Abbreviations: control (Ck); melatonin (M); salinity (S); drought (D); cadmium (Cd); cadmium + salinity (CdS); cadmium + drought (CdD); cadmium + salinity + drought (CdSD); cadmium + salinity + melatonin (CdSM); cadmium + drought + melatonin (CdDM); cadmium + drought + salinity + melatonin (CdDSM).

**Figure 6 ijms-25-08901-f006:**
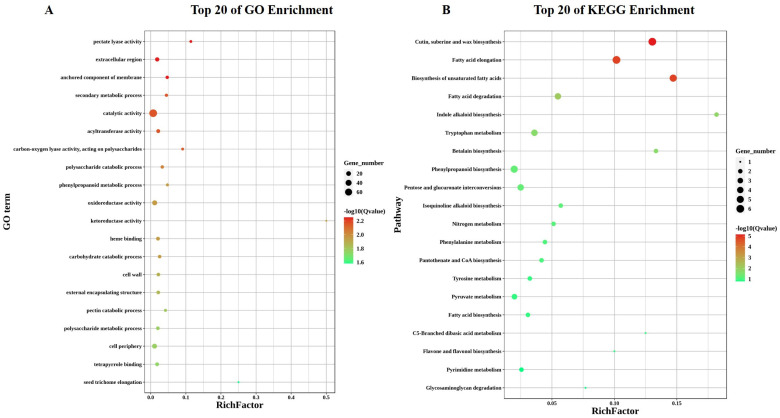
Analysis of differentially expressed genes (DEGs) in the salmon module based on gene ontology (GO) and Kyoto Encyclopedia of Genes and Genomes (KEGG). (**A**) The top 20 GO enrichments; (**B**) the top 20 KEGG enrichments. (Note: The q-value ranges from 0 to 0.05, with closer values indicating higher significance. The number of genes enriched in pathways is provided).

**Figure 7 ijms-25-08901-f007:**
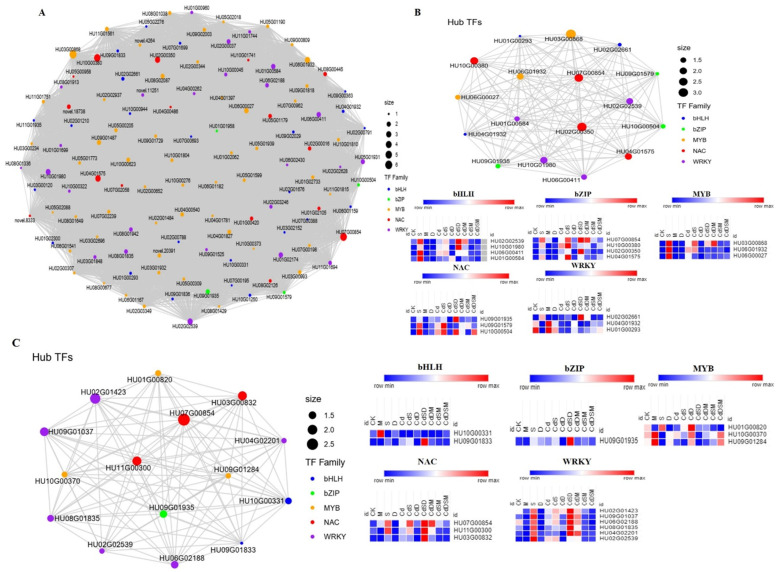
Identification and selection of hub transcription factors (TFs) in salmon and blue modules. (**A**) Network analysis of TFs in the salmon module; (**B**) network of top 17 hub TFs and related genes in salmon module; (**C**) network analysis of hub TFs in blue module. Abbreviations: control (Ck); melatonin (M); salinity (S); drought (DA); cadmium (Cd); cadmium + salinity (CdS); cadmium + drought (CdD); cadmium + salinity + drought (CdSD); cadmium + salinity + melatonin (CdSM); cadmium + drought + melatonin (CdDM); cadmium + drought + salinity + melatonin (CdDSM). WRKY TF (WRKY); myeloblastosis TF (MYB); basic-helix–loop-helix TF (bHLH); basic leucine zipper (bZIP); NAM, ATAF, and CUC (NAC).

**Figure 8 ijms-25-08901-f008:**
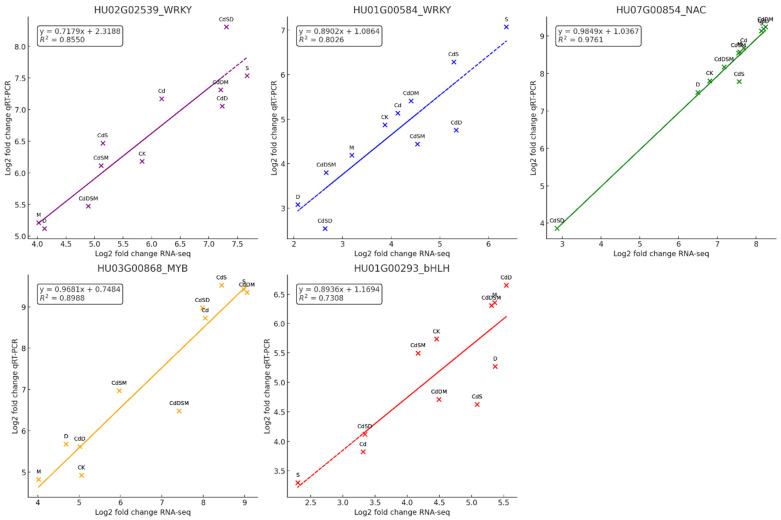
Gene expression validation of five selected genes under various treatments such as Ck, M, S, D, Cd, CdS, CdD, CdSD, CdSM, CdDM, and CdDSM. Each plot shows a strong correlation with respective regression equations and R-squared values: *HU02G02539*_WRKY (y = 0.7173x + 2.1388, R^2^ = 0.8350), *HU01G05984*_WRKY (y = 0.9507x + 1.0864, R^2^ = 0.8026), *HU07G00854*_NAC (y = 0.9849x + 1.0367, R^2^ = 0.9711), *HU03G00886*_MYB (y = 0.9814x + 0.7484, R^2^ = 0.9545), and *HU01G00293*_bHLH (y = 0.9793x + 1.1094, R^2^ = 0.9277). The graph compares log2 fold changes in RNA-seq and qRT-PCR analyses for five genes. Abbreviations: control (Ck); melatonin (M); salinity (S); drought (D); cadmium (Cd); cadmium + salinity (CdS); cadmium + drought (CdD); cadmium + salinity + drought (CdSD); cadmium + salinity + melatonin (CdSM); cadmium + drought + melatonin (CdDM); cadmium + drought + salinity + melatonin (CdDSM).

## Data Availability

The data generated and analyzed in this study are available in the Appendix A.

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
