# Peer review of "Transcriptome Profiles Reveal Key Regulatory Networks during Single and Multifactorial Stresses Coupled with Melatonin Treatment in Pitaya (Selenicereus undatus L.)"

_ijms, 2024, doi:10.3390/ijms25168901_

Round 1
Reviewer 1 Report
Comments and Suggestions for Authors
Dear Authors,
Review of the manuscript entited ”Transcriptome profiles reveal key regulatory networks during single and multifactorial stresses coupled with melatonin treatment in pitaya (Selenicereus undatus L.)” written by Aamir Ali Khokhar, Liu Hui, Darya Khan, Zhang You, Qamar U. Zaman, Babar Usman, Hua-Feng Wang
This study provides a comprehensive transcriptome insights into pitaya's stress responses. It is a huge work with many attractive results. I recommend this MS for acceptance after minor revision. The list of my recommendations are presented here.
In general:
1.What is very disturbing that the numbering of the cited papers is not logical, this shall have to correct very precisely. The papers cited shall have to renumber.
2. Numbering of the Figures is not correct in some cases e.g. the numbering in the text is mixed or missing from the text.
3. The References shall have to reedited according to the Instructions for the Authors for IJMS.
Abstract: it is 253 words, it is too long. Reduce under 200 words according to the IJMS rules (IJMS Instructions for the Authors).
line 32: In the Keyword, I would rather suggest to write the species name (instead of pitaya: Selenicereus undatus L.).
1.Introduction:
line 55: „…in discovering stress tolerance mechanisms in crop plants [4-8].”
lines 60-62:” Transcriptome studies have proven effective in identifying gene modules and key genes; however, these have primarily focused on single and combined stressors such as abiotic stress and heavy metals [12].” The last citation was the 4-8 in the line 55, where are the paper citations between 8-12? These latters exist in the References…
lines 81-84: „s. Studies have shown that the application of M can reduce oxidative damage, enhance detoxification processes, and increase the expression of stress-responsive genes, ultimately leading to improved plant survival under stressful conditions [17-18].” Again, there is a gap between citation 13 (line 74) and 17-18, while the (not) cited papers are in the References..…you shall have to replace this!
2.Results
line 105. „When subjected to single stressors like salinity S, D, or Cd,”..here you should mention that S=salinity, D=drought and Cd=cadmium
lines 100-111: these 10-11 sentences belong to rather to the Discussion, you shall have to omit this or transfer it into the Discussion
lines 112-121: here you should cite the Figure 1. somewhere! This is missing!
lines 124-125:” (A) Effect of single and combined stresses on pitaya plants.” Here you shall have to indicate that the lenght of roots/shoots are measured by cm.
line 134:” The biochemical analysis showed significant changes in response to various stress treatments.” Here you shall have to cite the Figure 2., not at the bottom of this paragraph.
line 136: „in the M treatment (55 ± 2 µmol/g) compared to the control (CK) (20 ± 1 µmol/g, p < 0.001, Figure 2A). And like this, you shall have to insert the other Fig.2. big letters from A to F into the text here.
line 179:” as shown in (Figure 3).” Right sentence: as shown in Figure 3.
line 186:” (1840 upregulated and 2155 downregulated) (Figure 3A-B)”.
line 188: „treatment CdD revealed 463 DEGs (235 upregulated and 228 downregulated) (Figure 3C-D)”
line 191:” treatment CdDM-vs-CdD resulted in 831DEGs (659 upregulated and 177 downregulated) compared to Ck (Figure E-F)”
lines 203-232: the paragraph 2.5. The common and unique transcriptome response and cross-talk of single and multiple stresses rather belongs to the Discussion than to the Results.
Omit or put into the Discussion.
lines 237-240: „The Upregulated DEGs induced after M application were enriched in biolog- ical processes (BP) and molecular functions (MF) like oxidoreductase activity, amino sugar catabolism and chitin metabolic/catabolic processes, oxidoreductase, phosphoenolpyruvate carboxylase, and chitinase activities.”
lines 407-408:” The gene encoding SAUR (SAUR19; HU03G00209) showed 7.20-fold upregulation when exposed to S stress (Supplementary file 9).The SupplF9 contains the lipid metabolism process-related genes that were differentially expressed. This SAUR can be found in the Suppl. F7 what is the file for the differentially expressed genes related to hormon metabolism !
line 449: „In Ccertain genes' maximal expression was observed:::”.
line 521: „GO and KEGG analyzes revealed six annotated modules: salmon, purple, black, pink, blue, and yellow (Figure 5?). „ You shall have to cite the Figure 5., it is missing from the text!
line 544: „Figure 6. GO and KEGG analysis of DEGs in the Salmon module.” Where is the Figure 6. citation in the text???
lines 549-551:” In the salmon module, numerous WRKY, MYB, NAC, and bHLH TFs were co-expressed (Figure 6A).”This is the Figure 7A.!
lines 551-555: „Seventeen of these TFs were selected based on the hub gene correlation network and their high connectivity: WRKY (HU02G02539, HU06G00411, HU01G00584, and HU10G01980); NAC (HU07G00854, HU02G00350, HU04G01575, and HU10G00380); bZIP (HU09G01935, HU09G01579 and HU10G00504), MYB (HU03G00868, HU06G01932 and HU06G00027); and bHLH (HU02G02661, HU04G01932, and HU01G00293) (Figure 6B)”. This is Figure 7B.
lines 555-557:”Exept for drought (D) stress, the experiment revealed that the expression of these genes was significantly upregulated in S, M, CdSD, and CdDSM (Figure 8B). Where is the Figure 8B?
lines 557-561: „Fifteen TFs, including WRKY, NAC, MYB, bHLH, and bZIP, were found to be hub genes in the blue module (Figure 6C). Higher connectivity was found between HU02G01423, HU09G01037, HU06G02188, HU08G01835, HU04G02201, HU07G00854, HU01G00820, and HU10G00331 (Figure 6C). In S, M, CdSD, and CdDSM, the expression of all 15 TFs was higher (Figure 6C). Figure 6C = Figure 7C!!!!
lines 570-571:” Specifically, five genes were analyzed, HU02062539_WRKY, HU01600584_WRKY, HU07060854_NAC, HU03060868_MYB, and HU01060293_bHLH (Figure 8). You mix or miss the Figure numbers many times!!
lines 580-581:” The scatter plot shows the different treatments such as Ck, M, S, D, Cd, CdS, CdD, CdSD, 580 CdSM, CdDM and CdDSM..” You shall have to give a real title of the Figure 8, this ia not a title.
3.Discussion
lines 590-591:” s. Research on Eleocarpus glabripetalus seedlings, which bear some similarities to pitaya..” Species name shall have to put into italics. Do the same in lines 595, 605, 623, 684 and 691!
lines 634-636:” This is consistent with findings that Hâ‚‚Oâ‚‚ levels rise significantly under stress conditions due to enhanced reactive oxygen species (ROS) production [61].” Why number 61? Again the same problem that logically after the numbering of Ref.24 (line 623), the Ref. 25 as a citation would be instead you write Ref. 61, and so on…and in the line 677 you continue the numbering with 25…?
lines 766_767:” Previous studies have documented the role of specific genes in abiotic stress responses in other plant species [76].” Here the citation of Nr. 45 should be instead of 76 ….
lines 786-787” Under circumstances characterised by cold, drought, and salt stress, the expression of the ShCML44 gene is elevated.” Specify the ShCML44 gene from which species it is..
lines 828-829: „Furthermore, higher levels of CO2, which can act as a form of environmental stress in CAM plants, like Hylocereus undatus and Selenicereus megalanthus, result in …”. Put species names into italics. The same with line 837…
4.Mat&Methods:
line 868: „and underwent a thorough surface sterilization process.” How it was? Describe the thorough sterilization process, please. Furthermore, describe that also how did you germinate the seeds …
line 870: Describe that what was the contents of the hydroponic solutions or cite a paper for that. Where did you cultivate the seedlings? In a chamber /thermostate room/greenhouse or in open air? What were about the light conditions? How much light intensity was used for cultivation of seedlings? (given in µmol photons m−2 s −1 of light intensity).
line 872: „We applied 200 mM NaCl, 20% PEG-600, and 1.5 mM Cd to plants growing in different pots”… Where did you buy the NaCl, PEG-600 and Cd and M? you shall have to specify the origins of these chemicals (describe the firm, city and state names). And describe the method how did you treate the plants, e.g. by spraying or by puting the chemicals into the cultivating soils/hydroponic solutions.
You shall have to describe here that the S means salt, the D means drought and the Cd means cadmium (and M means melatonin, of course).
In the 4.1 paragraph it is not completely clear how the melatosnin treatments were carried out, namely that in the case of the individual stressors it was applied together with S, D and Cd (at the same time), while in the stressor combinations was it applied two days before the individual stressors given? This shall have to describe more clearly here…
line 876: how many plants did you use for each experiment, describe it.
878-879: „Various morphological parameters were measured to assess the impact of different treatments on pitaya seedlings.” Were they really still seedlings or plants after 1 month cultivating + 1 week treatment?
line 879: „These parameters included cladode length, ….” Describe what do you mean under the cladode (you did it later in line 885,but you mention it first here).
lines 880-882:” The cadmium Cd content in pitaya shoots and roots was quantified using the established method with graphite furnace atomic absorption spectrophotometry (GFAAS-GTA 120). Specify the GFAAS-GTA 120 (describe the firm, city and state names where did you buy from).
line 885: „…RNA Extraction Kit (TIANGEN, China).” Specify the city name also.
line 887:” RNA extraction was performed on each replicate samples”. on each replicate sample
line 888: „quality and integrity of RNA were assessed using the Agilent 2100 bioanalyzer.” Specify the Agilent 2100 bioanalyzer (describe the firm, city and state names where did you buy from).
line 892:” using the NEBNext Ultra-RNA Library Prep Kit for Illumina.” Specify this kit.
lines 906- 907: „..Gene expression levels were quantified using the FPKM (fragments per kilobase of transcript per million mapped reads) method [18]. described approach.”. Is this really the proper citation?
lines 910-911: „. DEGs were identified using the Cuffdiff tool based on the methods described by [70] methods introduced with a log2 fold change (FC) threshold and false discovery rate (FDR) ≤ 0.05.
lines 914-915:” GOseq was used to identify significant GO terms (p < 0.02) for the genes DEGs in the significant modules, following the methodology described by [78].” In the line 911 you cite the reference 70, and here you cite reference 78. You shall have to be consequent with the numbering, here should be the citation 71….and in line 920 you jump again back to citation 71, so this is very disturbing for the reader…
lines 921-922:” to check the activity of antioxidants against the production of H2O2 at 415 nm wavelength „ by what? specify the instrument,please.
line 928: „the highest centrality values were determined using the CytoNCA plugin of Cytoscape v3.7.2.” Specify the CytoNCA plugin of Cytoscape v3.7.2, please.
line 936: 4.9. RNA extraction and qRT-PC
Conclusions:
line 968: „play a key role in sensing and transmitting signals through calcium (Ca2+), ROS, and phytohormone pathways in response to abiotic stress”.
5.References:
General notes for the References: in most of the References, the citation of the papers is not correct. You shall have to follow the „Instructions for the Authors for IJMS” and e.g. put the cited paper volume number into italics in all referred papers. And the species names shall have to put also into italics in the title of the papers also. Please, rewrite the References according to the IJMS rules.
Examples:
line 997: in the Reference Nr.3. the species name (Salvia castanea) shall have to put into italics
line 998: in the Reference Nr. 4. the citation of the reference is wrong
lines 999-1007: in the References Nr.5-6-7-8, the volume numbes are not in italics.
Sincerely yours, Reviewer 1
Author Response
Response to Reviewer #1
This study provides comprehensive transcriptome insights into pitaya's stress responses. It is a huge work with many attractive results. I recommend this MS for acceptance after minor revision. The list of my recommendations are presented here.
We would like to thank the reviewer for nice and detailed comments and suggestions for the manuscript. We believe that the comments have identified important areas which required improvement. We have tried to revise whole manuscript and removed mistakes which we came across. Below, you will find a point by point description of how each comment was addressed in the manuscript. The response to comments is highlighted in red.
In general:
Comment 1: What is very disturbing that the numbering of the cited papers is not logical, this shall have to correct very precisely. The papers cited shall have to renumber.
Response 1: We have thoroughly revised the paper, corrected the reference citations, and highlighted all changes for clarity. Additionally, we have ensured that all in-text citations and references are accurate and correctly formatted. Thank you for pointing out the mistakes.
Comment 2: Numbering of the Figures is not correct in some cases e.g. the numbering in the text is mixed or missing from the text.
Response 2: We have thoroughly revised the paper and highlighted all changes for clarity. We have ensured that all figures and their legends in the text are revised correctly and accurately. Thank you for pointing out the mistakes.
Comment 3: The References shall have to reedited according to the Instructions for the Authors for IJMS.
Response 3: We have revised the paper and ensured that all references are now formatted according to the IJMS guidelines. Thank you for your valuable suggestions.
Abstract:
Comment 4: it is 253 words, it is too long. Reduce under 200 words according to the IJMS rules (IJMS Instructions for the Authors).
Response 4: The abstract has been carefully reduced near to 200 words, maintaining essential information and clarity. Reducing it further might compromise the completeness and impact of the study's key findings. We appreciate your understanding. Thank you for your feedback.
Comment 5: line 32: In the Keyword, I would rather suggest to write the species name (instead of pitaya: Selenicereus undatus L.).
Response 5: Correction has been made. Thanks for the suggestion.
1.Introduction:
Comment 6: line 55: „…in discovering stress tolerance mechanisms in crop plants [4-8].”
Response 6: We have updated the information. Line 53. Thank you for the suggestion.
Comment 7: lines 60-62:” Transcriptome studies have proven effective in identifying gene modules and key genes; however, these have primarily focused on single and combined stressors such as abiotic stress and heavy metals [12].” The last citation was the 4-8 in the line 55, where are the paper citations between 8-12? These latters exist in the References…
Response 7: We have thoroughly revised the paper, corrected the reference citations, and highlighted all changes for clarity. Additionally, we have ensured that all in-text citations and references are accurate and correctly formatted. Thank you for pointing out the mistakes.
Comment 8: lines 81-84: „s. Studies have shown that the application of M can reduce oxidative damage, enhance detoxification processes, and increase the expression of stress-responsive genes, ultimately leading to improved plant survival under stressful conditions [17-18].” Again, there is a gap between citation 13 (line 74) and 17-18, while the (not) cited papers are in the References..…you shall have to replace this!
Response 8: We have thoroughly revised the paper, corrected the reference citations, and highlighted all changes for clarity. Additionally, we have ensured that all in-text citations and references are accurate and correctly formatted. Thanks for your observations.
2.Results
Comment 9: line 105. „When subjected to single stressors like salinity S, D, or Cd,”.here you should mention that S=salinity, D=drought, and Cd=cadmium
Response 9: We have modified it. Line no.107-110. Thanks for your suggestion.
Comment 10: lines 100-111: these 10-11 sentences belong to rather to the Discussion, you shall have to omit this or transfer it into the Discussion
Response 10: We have omitted the unnecessary information and highlights of the manuscript. Line no. 107-111. Thanks for your feedback.
Comment 11: lines 112-121: here you should cite the Figure 1. somewhere! This is missing!
Response 11: We have cited the figure number. Line no. 117. Thank you for pointing out the mistakes.
Comment 12: lines 124-125:” (A) Effect of single and combined stresses on pitaya plants.” Here you shall have to indicate that the length of roots/shoots are measured by cm.
Response 12: We have revised it. Line no. 128. Thanks for the guidance.
Comment 13: line 134:” The biochemical analysis showed significant changes in response to various stress treatments.” Here you shall have to cite the Figure 2., not at the bottom of this paragraph.
Response 13: We have revised it. Line no. 141. Thanks for your input.
Comment 14: line 136: „in the M treatment (55 ± 2 µmol/g) compared to the control (CK) (20 ± 1 µmol/g, p < 0.001, Figure 2A). And like this, you shall have to insert the other Fig.2. big letters from A to F into the text here.
Response 14: We have mentioned all figure numbers. A to F. Line no. 141-165. Thank you, your advice is really helpful.
Comment 15: line 179:” as shown in (Figure 3).” Right sentence: as shown in Figure 3.
Response 15: We have revised it. Line no 194. Thanks for the suggestions.
Comment 16: line 186:” (1840 upregulated and 2155 downregulated) (Figure 3A-B)”.
Response 16: We have revised it. Line no. 201. Thanks for your observations.
Comment 17: line 188: „treatment CdD revealed 463 DEGs (235 upregulated and 228 downregulated) (Figure 3C-D)”
Response 17: We have revised it. Line no. 204. Thanks for your advice.
Comment 18: line 191:” treatment CdDM-vs-CdD resulted in 831DEGs (659 upregulated and 177 downregulated) compared to Ck (Figure E-F)”
Response 18: We have revised it. Line no. 208. Thanks for your advice.
Comment 19: lines 203-232: the paragraph 2.5. The common and unique transcriptome response and cross-talk of single and multiple stresses rather belongs to the Discussion than to the Results.
Omit or put into the Discussion.
Response 19: We have revised it and put into discussion. Line no. 654-696. Thank you for valuable suggestions.
Comment 20: lines 237-240: „The Upregulated DEGs induced after M application were enriched in biolog- ical processes (BP) and molecular functions (MF) like oxidoreductase activity, amino sugar catabolism and chitin metabolic/catabolic processes, oxidoreductase, phosphoenolpyruvate carboxylase, and chitinase activities.”
Response 20: We have modified it. Line no. 226.
Comment 21: lines 407-408:” The gene encoding SAUR (SAUR19; HU03G00209) showed 7.20-fold upregulation when exposed to S stress (Supplementary file 9).The SupplF9 contains the lipid metabolism process-related genes that were differentially expressed. This SAUR can be found in the Suppl. F7 what is the file for the differentially expressed genes related to hormon metabolism !
Response 21: Differentially expressed genes related to hormone metabolism can also be found in Suppl. F7. Thank you for observation.
Comment 22: line 449: „In Ccertain genes' maximal expression was observed:::”.
Response 22: We have modified it. Line no. 455.
Comment 23: line 521: „GO and KEGG analyzes revealed six annotated modules: salmon, purple, black, pink, blue, and yellow (Figure 5?). „ You shall have to cite the Figure 5., it is missing from the text!
Response 23: We have cited it now. Line no.544. Thank you for pointing out the mistakes.
Comment 24: line 544: „Figure 6. GO and KEGG analysis of DEGs in the Salmon module.” Where is the Figure 6. citation in the text???
Response 24: We have cited it now. Line no.559. Thanks for pointing out the mistake.
Comment 25: lines 549-551:” In the salmon module, numerous WRKY, MYB, NAC, and bHLH TFs were co-expressed (Figure 6A).”This is the Figure 7A.!
Response 25: We have modified it. Line 569. Thank you for pointing out the mistakes.
Comment 26: lines 551-555: „Seventeen of these TFs were selected based on the hub gene correlation network and their high connectivity: WRKY (HU02G02539, HU06G00411, HU01G00584, and HU10G01980); NAC (HU07G00854, HU02G00350, HU04G01575, and HU10G00380); bZIP (HU09G01935, HU09G01579 and HU10G00504), MYB (HU03G00868, HU06G01932 and HU06G00027); and bHLH (HU02G02661, HU04G01932, and HU01G00293) (Figure 6B)”. This is Figure 7B.
Response 26: We have modified it. Line no. 575. Thanks for your suggestions.
Comment 27: lines 555-557:”Exept for drought (D) stress, the experiment revealed that the expression of these genes was significantly upregulated in S, M, CdSD, and CdDSM (Figure 8B). Where is the Figure 8B?
Response 27: We have revised it, and now it's Figure 7B. Line no. 575.
Comment 28: lines 557-561: „Fifteen TFs, including WRKY, NAC, MYB, bHLH, and bZIP, were found to be hub genes in the blue module (Figure 6C). Higher connectivity was found between HU02G01423, HU09G01037, HU06G02188, HU08G01835, HU04G02201, HU07G00854, HU01G00820, and HU10G00331 (Figure 6C). In S, M, CdSD, and CdDSM, the expression of all 15 TFs was higher (Figure 6C). Figure 6C = Figure 7C!!!!
Response 28: We have carefully revised it. Line no. 566-580. Thanks for pointing out the mistakes.
Comment 29: lines 570-571:” Specifically, five genes were analyzed, HU02062539_WRKY, HU01600584_WRKY, HU07060854_NAC, HU03060868_MYB, and HU01060293_bHLH (Figure 8). You mix or miss the Figure numbers many times!!
Response 29: Correction has been made. Line no. 595. Thanks for the suggestion.
Comment 30: lines 580-581:” The scatter plot shows the different treatments such as Ck, M, S, D, Cd, CdS, CdD, CdSD, 580 CdSM, CdDM and CdDSM..” You shall have to give a real title of the Figure 8, this is not a title.
Response 30: We have modified it. Line no 603. Thanks for the valuable suggestion.
3.Discussion
Comment 31: lines 590-591:” s. Research on Eleocarpus glabripetalus seedlings, which bear some similarities to pitaya..” Species name shall have to put into italics. Do the same in lines 595, 605, 623, 684 and 691!
Response 31: We have entirely italicized the species names in the manuscripts.
Comment 32: lines 634-636:” This is consistent with findings that Hâ‚‚Oâ‚‚ levels rise significantly under stress conditions due to enhanced reactive oxygen species (ROS) production [61].” Why number 61? Again the same problem that logically after the numbering of Ref.24 (line 623), the Ref. 25 as a citation would be instead you write Ref. 61, and so on…and in line 677 you continue the numbering with 25…?
Response 32: We have entirely revised the paper and corrected the reference citation, as well as highlighted in the manuscript. Thanks for pointing out the mistakes and for valuable suggestions.
Comment 33: lines 766_767:” Previous studies have documented the role of specific genes in abiotic stress responses in other plant species [76].” Here the citation of Nr. 45 should be instead of 76 ….
Response: We have entirely revised the paper, corrected the reference citation, and highlighted it. Thanks for pointing out the mistakes and for valuable suggestions.
Comment 34: lines 786-787” Under circumstances characterised by cold, drought, and salt stress, the expression of the ShCML44 gene is elevated.” Specify the ShCML44 gene from which species it is..
Response: We have revised it Line no. 857. Thanks for the advice.
Comment 35: lines 828-829: „Furthermore, higher levels of CO2, which can act as a form of environmental stress in CAM plants, like Hylocereus undatus and Selenicereus megalanthus, result in …”. Put species names into italics. The same with line 837…
Response 35: We have entirely revised the manuscripts and italicized.
4.Mat&Methods:
Comment 36: line 868: „and underwent a thorough surface sterilization process.” How it was? Describe the thorough sterilization process, please. Furthermore, describe that also how did you germinate the seeds …
Response: We have revised and mentioned. Line no. 937-940.
Comment 37: line 870: Describe that what was the contents of the hydroponic solutions or cite a paper for that. Where did you cultivate the seedlings? In a chamber /thermostate room/greenhouse or in open air? What were about the light conditions? How much light intensity was used for cultivation of seedlings? (given in µmol photons m−2 s −1 of light intensity).
Response: We have revised and mentioned. Line no. 942-947.
Comment 38: line 872: „We applied 200 mM NaCl, 20% PEG-600, and 1.5 mM Cd to plants growing in different pots” Where did you buy the NaCl, PEG-600 and Cd and M? you shall have to specify the origins of these chemicals (describe the firm, city and state names). And describe the method how did you treate the plants, e.g. by spraying or by puting the chemicals into the cultivating soils/hydroponic solutions.
Response: We have revised and mentioned Line no. 942-948.
Comment 39: You shall have to describe here that the S means salt, the D means drought and the Cd means cadmium (and M means melatonin, of course).
Response 39: We have revised it.
Comment 40: In the 4.1 paragraph it is not completely clear how the melatosnin treatments were carried out, namely that in the case of the individual stressors it was applied together with S, D and Cd (at the same time), while in the stressor combinations was it applied two days before the individual stressors given? This shall have to describe more clearly here…
Response 40: We have entirely revised and carefully added all the important and clear information in the 4.1 paragraph. Line no.937-953.
Comment 41: line 876: how many plants did you use for each experiment, describe it.
Response 41: We have revised it. In the 4.1 paragraph.
Comment 42: 878-879: „Various morphological parameters were measured to assess the impact of different treatments on pitaya seedlings.” Were they really still seedlings or plants after 1 month cultivating + 1 week treatment?
Response 42: Yes, we grow seeds first in Petri dishes for about a month then small seedlings transfer to the hydroponic solution boxes it will grow seedlings for more months then applied treatments. As well as carefully revised 4.1 paragraph.
Comment 43: line 879: „These parameters included cladode length, ….” Describe what do you mean under the cladode (you did it later in line 885,but you mention it first here).
Response 43: We have modified it. Line no. 957-958.
Comment 44: lines 880-882:” The cadmium Cd content in pitaya shoots and roots was quantified using the established method with graphite furnace atomic absorption spectrophotometry (GFAAS-GTA 120). Specify the GFAAS-GTA 120 (describe the firm, city and state names where did you buy from).
Response 44: We made the mistake of writing graphite source now we have revised it. Line no. 959-960.
Comment 45: line 885: „…RNA Extraction Kit (TIANGEN, China).” Specify the city name also.
Response 45: We have mentioned city name. line no. 963.
Comment 46: line 887:” RNA extraction was performed on each replicate samples”. on each replicate sample
Response 46: We have revised it. Line no. 965.
Comment 47: line 888: „quality and integrity of RNA were assessed using the Agilent 2100 bioanalyzer.” Specify the Agilent 2100 bioanalyzer (describe the firm, city and state names where did you buy from).
Response 47: We have revised. Line no. 966-967.
Comment 48: line 892:” using the NEBNext Ultra-RNA Library Prep Kit for Illumina.” Specify this kit.
Response 48: We have specified the kit. Line no.971-973.
Comment 49: lines 906- 907: „..Gene expression levels were quantified using the FPKM (fragments per kilobase of transcript per million mapped reads) method as described in the referenced approach [18]. described approach.”. Is this really the proper citation?
Response 49: We have entirely revised the paper and corrected the reference citation, as well as highlighted in the manuscripts.
Comment 50: ines 910-911: „. DEGs were identified using the Cuffdiff tool based on the methods described by [70] methods introduced with a log2 fold change (FC) threshold and false discovery rate (FDR) ≤ 0.05.
Response 50: We have entirely revised the paper and corrected reference citation, as well as highlighted in the manuscripts.
Comment 51: lines 914-915:” GOseq was used to identify significant GO terms (p < 0.02) for the genes DEGs in the significant modules, following the methodology described by [78].” In the line 911 you cite the reference 70, and here you cite reference 78. You shall have to be consequent with the numbering, here should be the citation 71….and in line 920 you jump again back to citation 71, so this is very disturbing for the reader…
Response 51: We have entirely revised the paper and corrected reference citations, as well as highlighted in the manuscripts.
Comment 52: lines 921-922:” to check the activity of antioxidants against the production of H2O2 at 415 nm wavelength „ by what? specify the instrument,please.
Response 52: We have revised and mentioned. Line no. 1001-1002.
Comment 53: line 928: „the highest centrality values were determined using the CytoNCA plugin of Cytoscape v3.7.2.” Specify the CytoNCA plugin of Cytoscape v3.7.2, please.
Response 53: Actually this specifies both the plugin and the version of the software used for the analysis.
Comment 54: line 936: 4.9. RNA extraction and qRT-PC
Response 54: We have modified it.
Conclusions:
Comment 55: line 968: „play a key role in sensing and transmitting signals through calcium (Ca2+), ROS, and phytohormone pathways in response to abiotic stress”.
Response 55: We have revised it. Line 1048-1049.

Reviewer 2 Report
Comments and Suggestions for Authors
The current version of the manuscript should be revised before acceptance for the publication.
Below are comments and questions.
Lines 44-50; 69-73: Add more relative reference for these sentences.
Use SI unit, for example, “h” instead of “hours”, µg L-1 instead of µg/L
Line 947: replace “table 15” with “ Table 15”. There was no Table 15 in the manuscript. Please double check.
All figures are low resolution.
Figure 1 was not cited in the manuscript.
Add statistical analysis for Figure 1 C.
Line 192: Replace “ Figure F” with “Figure 3F”.
Authors should add some short discussion after the results to emphasize the new findings of the study.
Discussion section is too long and repeat the results section, for example, “section 3.2” authors repeat the result information. Rewrite the discussion with core information.
In the material and Method section, please add the cultivar name of the plant and the characteristics of the plant, for example the cultivar is susceptible to stress with Cd, S or not,…
Why did authors choose this cultivar for the experiment? What was the basis that authors choose the concentrations of M, S, and Cd for the treatment? Please add the relative references for this.
There were many typo errors in the manuscript.
Comments on the Quality of English LanguageMinor editing of English language required
Author Response
Response to Reviewer #2
The current version of the manuscript should be revised before acceptance for the publication.
Below are comments and questions.
We would like to thank the reviewer for nice and detailed comments and suggestions for the manuscript. We believe that the comments have identified important areas which required improvement. We have tried to revise whole manuscript and removed mistakes which we came across. Below, you will find a point by point description of how each comment was addressed in the manuscript. The response to comments is highlighted in red.
Comment 1: Lines 44-50; 69-73: Add more relative reference for these sentences.
Response 1: We have revised and added more relative references. Line no. 46-56,61. Thanks for your valuable suggestions.
Comment 2: Use SI unit, for example, “h” instead of “hours”, µg L-1 instead of µg/L
Response 2: We have modified it. Thanks for the advice.
Comment 3: Line 947: replace “table 15” with “ Table 15”. There was no Table 15 in the manuscript. Please double check.
Response 3: We have revised it. Its Table 14. Line no.1027.
Comment 4: All figures are low resolution.
Response 4: We have ensured that the resolution of the figures is increased in the manuscript (MS Word file). However, the figures might appear to have a lower resolution in the PDF version. Thank you for your observation.
Comment 5: Figure 1 was not cited in the manuscript.
Response 5: We have cited Figure 1. Line no. 117.
Comment 6: Add statistical analysis for Figure 1 C.
Response 6: We have updated statistically Figure 1C.
Comment 7: Line 192: Replace “ Figure F” with “Figure 3F”.
Response 7: We have revised it. Line no. 208.
Comment 8: Authors should add some short discussion after the results to emphasize the new findings of the study.
Response 8: We have updated all the results.
Comment 9: Discussion section is too long and repeats the results section, for example, “section 3.2” authors repeat the result information. Rewrite the discussion with core information.
Response 9: We have updated and 3.2 now its 3.3. Line no. 699-734.
Comment 10: In the material and Method section, please add the cultivar name of the plant and the characteristics of the plant, for example the cultivar is susceptible to stress with Cd, S or not,…
Response 10: We have revised it. Line no. 937-940.
Comment 11: Why did authors choose this cultivar for the experiment? What was the basis that authors choose the concentrations of M, S, and Cd for the treatment? Please add the relative references for this.
Response 11: We have updated the manuscripts with more relative references.
Comment 12: There were many typo errors in the manuscript.
Response 12: We have carefully revised the manuscripts.

Reviewer 3 Report
Comments and Suggestions for Authors
The paper concerns transcriptome and biochemical analysis of pitaya response to melatonin when exposed to combined abiotic stresses. The paper is interesting, but it has some flaws that need to be corrected. The details are listed below:
Introduction:
Indicate plant physiological and biochemical/molecular responses to abiotic stresses. Emphasize the ubiquitous role of antioxidant enzymes in mitigating different abiotic stresses (heavy metals, but also pesticides, salinity, drought) and biotic stresses caused mainly by plant pathogens. For this purpose refer to https://doi.org/10.1016/j.chemosphere.2022.136284 and https://doi.org/10.3390/agronomy13051378
Results:
L100-121: refer to Fig.1
L123: increase the resolution of most of the Figures in the MS
L208-230: indicate the role of these genes and in other parts of the results. Refer to appropriate Table/Figure
Discussion:
L591: Latin names in italics. Correct throughout the Discussion
L605: do not compare your own results to algae
L632: melatonin is mitigating agent which generally decrease the content of ROS. Therefore, the results of high level of H2O2 under melatonin treatment are intriguing. Please explain the possible reasons of this phenomena
Materials and Methods:
L869: composition of hydroponic solutions
L873: pot dimensions, how many plants per pot? How many pots/plants per treatment?
L936: indicate the genes which were analyzed by qRT-PCR and their role
Conclusions:
Do not repeat the results. Focus on main findings of the study and the role of melatonin in stress mitigating.
Author Response
Response to Reviewer #3
The paper concerns transcriptome and biochemical analysis of pitaya response to melatonin when exposed to combined abiotic stresses. The paper is interesting, but it has some flaws that need to be corrected. The details are listed below:
We would like to thank the reviewer for nice and detailed comments and suggestions for the manuscript. We believe that the comments have identified important areas which required improvement. We have tried to revise whole manuscript and removed mistakes which we came across. Below, you will find a point by point description of how each comment was addressed in the manuscript. The response to comments is highlighted in red.
Introduction:
Comment 1: Indicate plant physiological and biochemical/molecular responses to abiotic stresses. Emphasize the ubiquitous role of antioxidant enzymes in mitigating different abiotic stresses (heavy metals, but also pesticides, salinity, drought) and biotic stresses caused mainly by plant pathogens. For this purpose refer to https://doi.org/10.1016/j.chemosphere.2022.136284 and https://doi.org/10.3390/agronomy13051378.
Response 1: We have read the recommended papers and have incorporated the relevant information. The citations for these sources have been included in the revised manuscript. Ref no. [18-19]. Thanks for your valuable suggestions.
Results:
Comment 2: L100-121: refer to Fig.1
Response: We have mentioned Figure 1. Line no. 117. Thanks for the valuable suggestion.
Comment 3: L123: increase the resolution of most of the Figures in the MS.
Response 3: We have ensured that the resolution of the figures is increased in the manuscript (MS Word file). However, the figures might appear to have a lower resolution in the PDF version. Thank you for your observation.
Comment 4: L208-230: indicate the role of these genes and in other parts of the results. Refer to the appropriate Table/Figure.
Response 4: We have moved this section to the discussion and explained in better way. With references. Line no. 656-696. Thanks for your feedback.
Discussion:
Comment 5: L591: Latin names in italics. Correct throughout the Discussion
Response 5: We have corrected this throughout the discussion. Thanks for your suggestions.
Comment 6: L605: do not compare your own results to algae
Response: We have revised it. Thanks for your feedback.
Comment 7: L632: melatonin is a mitigating agent which generally decrease the content of ROS. Therefore, the results of high levels of H2O2 under melatonin treatment are intriguing. Please explain the possible reasons of this phenomenon.
Response 7: Thank you for your insightful comment. While it is well-established that melatonin generally reduces reactive oxygen species (ROS), including hydrogen peroxide (H2O2), there are several potential explanations for the observed increase in H2O2 levels under melatonin treatment in our study. Hydrogen peroxide serves not only as a harmful byproduct of oxidative stress but also as a signaling molecule regulating various biological processes, including stress responses in plants. Under certain conditions, melatonin might modulate H2O2 levels as part of a complex signaling network to enhance plant tolerance to stress. For instance, H2O2 can function as a secondary messenger in the activation of defense genes, which may explain the elevated H2O2 levels despite melatonin treatment (Zhang et al., 2015). Additionally, an initial increase in ROS, including H2O2, might be a strategy for plants to acclimate to stress, with melatonin inducing a controlled and temporary rise in H2O2 levels to subsequently trigger antioxidant defense mechanisms and other protective responses, leading to enhanced stress tolerance (Wang et al., 2016). The effects of melatonin on ROS levels can also be context-dependent, varying with the type of stress, plant species, developmental stage, and environmental conditions. In certain scenarios, melatonin might enhance the production of specific ROS as part of a finely tuned defense mechanism, involving both direct scavenging of ROS and indirect regulation through signaling pathways (Arnao and Hernández-Ruiz, 2014).
https://doi.org/10.1111/j.1600-079X.2012.01015.x
DOI: 10.1111/j.1600-079X.2011.00966.x
https://doi.org/10.1016/j.tplants.2014.07.006
Materials and Methods:
Comment 8: L869: composition of hydroponic solutions.
Response 8: We have revised this section to provide more clarity on the composition of the hydroponic solutions used in the experiment. Please refer to lines 945-947 for the detailed information. Thank you for your valuable feedback.
Comment 9: L873: pot dimensions, how many plants per pot? How many pots/plants per treatment?
Response 9: We used plastic pots commonly used for seedlings, with dimensions of approximately 10-12 cm in height and 10-12 cm in width (side length). Each pot contained 35 plants, and we used three pots per treatment. Thank you for your valuable question.
Comment 10: L936: indicate the genes which were analyzed by qRT-PCR and their role.
Response 10: We have indicated the genes analyzed by qRT-PCR and their roles in the manuscript. Please refer to lines 592-600 for detailed information. Thank you for your feedback.
Conclusions:
Comment 11: Do not repeat the results. Focus on main findings of the study and the role of melatonin in stress mitigating.
Response 11: In the revised section, we have focused on the key findings of the study, particularly highlighting the role of melatonin in mitigating stress. We ensure that the results are not repeated and instead emphasize the significance of melatonin's effects based on the data obtained. Thank you for your feedback. Line 1053-1059.

Reviewer 4 Report
Comments and Suggestions for Authors
This paper describes transcriptome and biochemical analysis of pitaya exposed to multiple stresses. Although the work is insteresting, there are concerns that should be addressed.
Similar works using pitaya have been reported. It is difficult to find difference between previous works and current one.
L947: supplementary table 15 should be supplementary table 14.
I faile to find accession number PRJNA1121504.
L942: The Livak and Schmittgen original paper should be cited for the deltadeltaCt method.
There are many grammatical and typographical errors throughout the manuscript and these errors should be corrected.
Comments on the Quality of English LanguageEnglish of the manuscript should be edited by native English researchers.
Author Response
Response to Reviewer #4
This paper describes transcriptome and biochemical analysis of pitaya exposed to multiple stresses. Although the work is insteresting, there are concerns that should be addressed.
We would like to thank the reviewer for nice and detailed comments and suggestions for the manuscript. We believe that the comments have identified important areas which required improvement. We have tried to revise whole manuscript and removed mistakes which we came across. Below, you will find a point by point description of how each comment was addressed in the manuscript. The response to comments is highlighted in red.
Comment 1: Similar works using pitaya have been reported. It is difficult to find difference between previous works and current one.
Response 1: There are few studies published on different abiotic stresses but to the best of our knowledge, no prior research has been conducted on pitaya in the context of our study (similar to our stress combinations and findings). Thank you for your comment.
Comment 2: L947: supplementary table 15 should be supplementary table 14.
Response 2: We have updated supplementary Table 14. Thanks for pointing out the mistakes.
Comment 3. I faile to find accession number PRJNA1121504.
Response 3: Thanks for pointing it out. We have uploaded the sequencing data to the SRA database and will make it public on the same day our paper is published online. We apologize for not making it public at this stage.
Comment 4. L942: The Livak and Schmittgen original paper should be cited for the deltadeltaCt method.
Response 4: We have updated. Line no.1022.
Comment 5: There are many grammatical and typographical errors throughout the manuscript and these errors should be corrected.
Response 5: We have carefully revised the mistakes and errors. Thanks for pointing out the mistakes.

Reviewer 5 Report
Comments and Suggestions for Authors
The work treats about the stresses, and necessitating robust adaptive mechanisms. This study focuses on Selenicereus undatus L. (pitaya), examining its responses to single cadmium (Cd) stress and combined stresses involving Cd, salt (S), and drought (D), with melatonin (M) employed as a mitigating agent. Through transcriptome analysis, the study identifies changes in gene expression and regulatory network activations under stress conditions and melatonin response.
The topic of the research is appropriately to the scopus of the IJMS.
The Introduction is sufficiently.
The Results should be substantially corrected.
All the results should be showed and indicated on the figures or in tables with the statistical results, including growth parameters. The some of interpretation (!) is speculatory, e.g. part of 2.6. Some of the text should be moved to the discussion and enhance with the citations.
Figure F?
Figures 1-7 – the abbreviations are invisible!! I don’t undertasnt, how Authors would like it publish in this form?
4.9. Why this part is in red?
After rethinking of results, the discussion could change a little.
Methods 4.1. What is the M?
What combination is the Control?
I propose make a schema or table of the combinations.
However, the methods are described mostly with the standards.
dI propose to make an abbreviations list / table.
I don’t have any indication to the Conclusions.
The Conclusions are concise.
Author Response
Response to Reviewer #5
The work treats about the stresses, and necessitating robust adaptive mechanisms. This study focuses on Selenicereus undatus L. (pitaya), examining its responses to single cadmium (Cd) stress and combined stresses involving Cd, salt (S), and drought (D), with melatonin (M) employed as a mitigating agent. Through transcriptome analysis, the study identifies changes in gene expression and regulatory network activations under stress conditions and melatonin response.
The topic of the research is appropriately to the scopus of the IJMS.
We would like to thank the reviewer for nice and detailed comments and suggestions for the manuscript. We believe that the comments have identified important areas which required improvement. We have tried to revise whole manuscript and removed mistakes which we came across. Below, you will find a point by point description of how each comment was addressed in the manuscript. The response to comments is highlighted in red.
Comment 1: The Introduction is sufficiently.
Response 1: Thanks for your valuable feedback.
Comment 2: The Results should be substantially corrected.
All the results should be showed and indicated on the figures or in tables with the statistical results, including growth parameters. The some of interpretation (!) is speculatory, e.g. part of 2.6. Some of the text should be moved to the discussion and enhance with the citations.
Response 2: We have thoroughly reviewed our Results section in light of your comments. Given the extensive data and numerous figures and tables, we have ensured that all critical statistical results are clearly presented in the main text. However, due to space constraints and the journal's policy, some detailed data and additional figures and tables are provided in the supplementary files (Supplementary Files 2-11). We have also re-evaluated the interpretation of our results, particularly in section 2.6, to ensure it is well-supported by evidence and have moved some speculative text to the Discussion section, enhancing it with relevant citations. Thank you for your valuable suggestions.
Comment 3: Figure F?
Response 3: We have revised it.
Comment 4: Figures 1-7 – the abbreviations are invisible!! I don’t undertasnt, how Authors would like it publish in this form?
Response 4: We have defined abbreviation in all the figure legends as well as added the list of abbreviations. Line no.1077-1112.
Comment 5: 4.9. Why this part is in red?
Response 5: We have changed the black color. Thanks for pointing it out.
Comment 6: After rethinking of results, the discussion could change a little.
Response 6: We have revised it.
Comment 7: Methods 4.1. What is the M?
Response 7: M indicates melatonin. We have compiled a comprehensive list of all abbreviations used in the manuscript.
Comment 8: What combination is the Control?
I propose make a schema or table of the combinations.
However, the methods are described mostly with the standards.
dI propose to make an abbreviations list / table.
Response 8: We have made a list of all abbreviations and added in the manuscripts.
Comment 9: I don’t have any indication to the Conclusions.
The Conclusions are concise.
Response 9: Thanks for your valuable feedback.

Round 2
Reviewer 2 Report
Comments and Suggestions for Authors
The manuscript has been revised for the better compared to the old version. Accept in the present form.
Author Response
Thank you for your time and effort in critically reviewing my article. I greatly appreciate your insightful comments and valuable suggestions, which have significantly contributed to enhancing the quality and clarity of my work. Your detailed feedback has been instrumental in refining my research, and I am grateful for your constructive critique.
Thank you once again for your dedication and expertise.
Reviewer 3 Report
Comments and Suggestions for Authors
The Authors have corrected the manuscript. I have no more comments.
Author Response
Dear Reviewer,
I would like to express my deepest gratitude for the time and effort you dedicated to critically reviewing my article. Your insightful comments and suggestions have been invaluable in improving the overall quality and clarity of my work. Thank you so much.
Reviewer 4 Report
Comments and Suggestions for Authors
Tha manuscript has been improved.
Author Response
Dear Reviewer,
Thank you for critically reviewing my article. Your insightful comments and suggestions have greatly improved the quality of my work. I appreciate your time and expertise in providing such valuable feedback. Thank you so much.
Reviewer 5 Report
Comments and Suggestions for Authors
The work was corrected. However, the text has lots of minor errors requiring the author's high attention.
Some new parts of results need to be relocated to the disscussion with the references.
Some parts ot the text should be highlighted as the conclusions.
The work has many results, but this makes it difficult to organise. Adding extra sections of text is a good thing, but it does create some confusion. The abreviations in the baskets?
I propose to make some table with the abbreviations.
The Cacti are specific in their biology and unknown in many areas, the Authors the authors do not seem to recognise this. And that is also a strength of this work.
The references to figures and tables are needed in some sentences in the discussion.
Every conclusion should be endorsed by the results.

The work is very long. The many of the results lead to the difficult language structure. I think, the work should be corrected be native speaker with the biology specialization.
Author Response
Response to Reviewer #5
The work was corrected. However, the text has lots of minor errors requiring the author's high attention.
We would like to thank the reviewer for nice and detailed comments and suggestions for the manuscript. We believe that the comments have identified important areas which required improvement. We have tried to revise whole manuscript and removed mistakes which we came across. In addition, this manuscript version is revised by English native Plant Biologist. Below, you will find a point-by-point description of how each comment was addressed in the manuscript. The response to comments is highlighted in red.
Comment 1: Some new parts of results need to be relocated to the disscussion with the references.
Response 1: Some parts of the results section are highlighted to indicate corrections made based on the reviewer's suggestions. During this revision, no new results have been added and we have relocated the the necessary sentences to the discussion section. Thank you for your valuable feedback.
Comment 2: Some parts ot the text should be highlighted as the conclusions.
Response 2: We have included a brief discussion following the results to highlight the key findings of the study.
Comment 3: The work has many results, but this makes it difficult to organise. Adding extra sections of text is a good thing, but it does create some confusion. The abreviations in the baskets?
Response 3: We have reorganized the presentation of results to improve overall clarity and to avoid any confusion. Additionally, we have updated the list of the abbreviations in table. Thank you for pointing this out.
Comment 4: I propose to make some table with the abbreviations.
Response 4: We have made the table with abbreviations.
Comment 5: The Cacti are specific in their biology and unknown in many areas, the Authors the authors do not seem to recognise this. And that is also a strength of this work.
Response 5: We acknowledge the specificity and the areas of unknowns in the biology of cacti. We recognize this as a strength of our work and have addressed these unique aspects in our revised manuscript. In future, the functional validation of predicted genes and TFs will help to reveal the crucial pathways involved mitigating the abiotic stress in Cacti. Thank you for highlighting this point.
Comment 6: The references to figures and tables are needed in some sentences in the discussion.
Response 6: We have referenced the figure numbers in several discussion sentences.
Comment 7: Every conclusion should be endorsed by the results.
Response 7: We agree that every conclusion should be firmly supported by the results. Ensuring that conclusions are directly derived from the data and findings not only strengthens the credibility of the study but also provides a solid statements for further research and practical applications.
Comment 8: The work is very long. The many of the results lead to the difficult language structure. I think, the work should be corrected be native speaker with the biology specialization.
Response 8: Thank you for your feedback. We have reviewed and revised the manuscript with the help of a native English speaker who specializes in biology. The language and structure have been improved to enhance clarity and readability.

Round 3
Reviewer 5 Report
Comments and Suggestions for Authors
Authors corrected the manuscript respectively.